# Simulating human impacts on global water resources using VIC-5

Bram Droppers[1], Wietse H.P. Franssen[1], Michelle T.H. van Vliet[2], Bart Nijssen[3], Fulco Ludwig[1]

[1] Water Systems and Global Change Group, Department of Environmental Sciences, Wageningen University, Wageningen, 6708 PB, The Netherlands
[2] Department of Physical Geography, Utrecht University, Utrecht, 3584 CS, The Netherlands
[3] Computational Hydrology Group, Department of Civil and Environmental Engineering, University of Washington, Seattle, 98195, United States of America

*Correspondence to:* Bram Droppers (bram.droppers@wur.nl)

**Abstract.** Questions related to historical and future water resources and scarcity have been addressed by several macro-scale hydrological models. One of these models is the Variable Infiltration Capacity (VIC) model. However, further model developments were needed to holistically assess anthropogenic impacts on global water resources using VIC.

Our study developed VIC-WUR, which extends the VIC model with: (1) integrated routing, (2) surface and groundwater use for various sectors (irrigation, domestic, industrial, energy, and livestock), (3) environmental flow requirements for both surface and groundwater systems, and (4) dam operation. Global gridded datasets on sectoral demands were developed separately and used as an input to the VIC-WUR model.

Simulated national water withdrawals were in line with reported FAO national annual withdrawals (adjusted $R^2 > 0.8$), both per sector as well as per source. However, trends in time for domestic and industrial water withdrawal were mixed compared to other previous studies. GRACE monthly terrestrial water storage anomalies were well represented (global mean RMSE of 1.9 mm and 3.5 mm for annual and interannual anomalies respectively), while groundwater depletion trends were overestimated. The implemented human impact modules increased simulated streamflow performance for 370 out of 462 human-impacted GRDC monitoring stations, mostly due to the effects of reservoir operation. An assessment of environmental flow requirements indicates that global water withdrawals have to be severely limited (by 39 %) to protect aquatic ecosystems, especially for groundwater withdrawals.

VIC-WUR has potential for studying impacts of climate change and anthropogenic developments on current and future water resources and sectoral specific-water scarcity. The additions presented here make the VIC model more suited for fully-integrated worldwide water-resource assessments.

## 1    Introduction

Questions related to historical and future water resources and scarcity have been addressed by several macro-scale hydrological models over the last few decades (Liang et al., 1994; Alcamo et al., 1997; Hagemann and Gates, 2001; Takata et al., 2003; Krinner et al., 2005; Bondeau et al., 2007; Hanasaki et al., 2008b; van Beek and Bierkens, 2009; Best et al., 2011). Early efforts focussed on the simulation of natural water resources and the impacts of land cover and climate change on water availability (Oki et al., 1995; Nijssen et al., 2001a; Nijssen et al., 2001b). Recently, a larger focus has been on incorporating anthropogenic impacts, such as water withdrawals and dam operations, into water resource assessments (Alcamo et al., 2003; Haddeland et al., 2006b; Biemans et al., 2011; Wada et al., 2011b; Hanasaki et al., 2018).

Global water withdrawals increased eight-fold over the last century and are projected to increase further (Shiklomanov, 2000; Wada et al., 2011a). Although water withdrawals are only a small fraction of the total global runoff (Oki and Kanae, 2006), water scarcity can be severe due to the variability of water in both time and space (Postel et al., 1996). Already severe water scarcity is experienced by two-thirds of the global population for at least part of the year (Mekonnen and Hoekstra, 2016). To stabilize water availability for different sectors (e.g. irrigation, hydropower, and domestic uses) dams and reservoirs were built, which are able to strongly affect global river streamflow (Nilsson et al., 2005; Grill et al., 2019). In addition, groundwater resources are being extensively exploited to meet increasing water demands (Rodell et al., 2009; Famiglietti, 2014).

One of widely-used macro-scale hydrological models is the Variable Infiltration Capacity (VIC) model. The model was originally developed as a land-surface model (Liang et al., 1994), but has been mostly used as a stand-alone hydrological model (Abdulla et al., 1996; Nijssen et al., 1997) using an offline routing module (Lohmann et al., 1996; Lohmann et al., 1998b, a). Where land-surface models focus on the vertical exchange of water and energy between the land surface and the atmosphere, hydrological models focus on the lateral movement and availability of water. By combining these two approaches, VIC simulations are strongly process-based and this, in turn, provides a good basis for climate-impact modelling.

VIC has been used extensively in studies ranging from: coupled regional climate model simulations
(Zhu et al., 2009; Hamman et al., 2016), combined river streamflow and water-temperature simulations
(van Vliet et al., 2016), hydrological sensitivity to climate change (Hamlet and Lettenmaier, 1999;
Nijssen et al., 2001a; Chegwidden et al., 2019), global streamflow simulations (Nijssen et al., 2001b),
sensitivity in flow regulation and redistribution (Voisin et al., 2018; Zhou et al., 2018), and real-time
drought forecasting (Wood and Lettenmaier, 2006; Mo, 2008). Several studies used VIC to simulate the
anthropogenic impacts of irrigation and dam operation on water resources (Haddeland et al., 2006a;
Haddeland et al., 2006b; Zhou et al., 2015; Zhou et al., 2016) based on the model setup of Haddeland et
al. (2006b). However, further developments were needed to holistically assess anthropogenic impacts
on global water resources using VIC (Nazemi and Wheater, 2015a, b; Döll et al., 2016; Pokhrel et al.,

69    2016).

Firstly, the VIC model did not yet include groundwater withdrawals or water withdrawals from
domestic, manufacturing, and energy (thermoelectric) sources. Although these sectors use less water
than irrigation (Shiklomanov, 2000; Grobicki et al., 2005; Hejazi et al., 2014) they are locally important
actors (Gleick et al., 2013), especially for the water-food-energy nexus (Bazilian et al., 2011). Sufficient
water supply and availability are essential for meeting a range of local and global sustainable
development goals related to water, food, energy, and ecosystems (Bijl et al., 2018). Secondly,
environmental flow requirements (EFRs) were often neglected (Pastor et al., 2014), even though they
are "necessary to sustain aquatic ecosystems which, in turn, support human cultures, economies,
sustainable livelihoods, and well-being" (Brisbane Declaration, 2017). Anthropogenic alterations
already strongly affect freshwater ecosystems (Carpenter et al., 2011), with more than a quarter of all
global rivers experiencing very high biodiversity threats (Vorosmarty et al., 2010). By neglecting EFRs,
sustainable water availability for anthropogenic uses is overestimated (Gerten et al., 2013). Lastly, while
the model setup of Haddeland et al. (2006b) already included important anthropogenic impact modules
(i.e. irrigation and dam operation), these were not fully integrated yet. Therefore multiple successive
model runs were required (see Sect. 2.1) which was computationally expensive, especially for global
water resources assessments.
Recently version 5 of the VIC model (VIC-5) was released (Hamman et al., 2018), which focussed on
improving the VIC model infrastructure. These improvements provide the opportunity to fully integrate
human-impacts into the VIC model framework, while reducing computation times. Here the newly
developed VIC-WUR model is presented (named after the developing team at Wageningen University
and Research). The VIC-WUR model extends the existing VIC-5 model with several modules that
simulate the anthropogenic impacts on water resources. These modules will implement previous major
works on anthropogenic impact modelling as well as integrate environmental flow requirements into
VIC-5. The modules include: (1) integrated routing, (2) surface and groundwater use for various sectors
(irrigation, domestic, industrial, energy and livestock), (3) environmental flow requirements for both
surface and groundwater systems, and (4) dam operation.
The next section first describes the original VIC-5 hydrological model (Sect. 2.1), which calculates
natural water resource availability. Subsequently the integration of the anthropogenic impact modules,
which modify the water resource availability, are described (Sect. 2.2). Global anthropogenic water uses
for each sector are also estimated (Sect. 2.3). To assess the capability of the newly developed modules,
the VIC-WUR results were compared with FAO national water withdrawals by sector and by source
(FAO, 2016); Huang et al. (2018), Steinfeld et al. (2006), and Shiklomanov (2000) data on water
withdrawals by sector; GRACE terrestrial water storage anomalies (NASA, 2002); GRDC streamflow
timeseries (GRDC, 2003); and Yassin et al. (2019) and Hanasaki et al. (2006) data on reservoir operation
(Sect. 3.2). VIC-WUR simulations results are also compared with various other state-of-the-art global
hydrological models. Lastly, the impacts of adhering to surface and groundwater environmental flow
requirements on water availability are assessed (Sect. 3.3). This assessment is included to indicate the
effects of the newly integrated surface and groundwater environmental flow requirements on worldwide
water availability.

## 2    Model development

### 2.1    VIC hydrological model

The basis of the VIC-WUR model is the Variable Infiltration Capacity model version 5 (VIC-5) (Liang et al., 1994; Hamman et al., 2018). VIC-5 is an open source macro-scale hydrological model that simulates the full water and energy balance on a (latitude – longitude) grid. Each grid cell accounts for sub-grid variability in land cover and topography, and allows for variable saturation across the grid cell. For each sub-grid the water and energy balance is computed individually (i.e. sub-grid do not exchange water or energy between one another). The methods used to calculate the water and energy balance are summarized in Appendix A, mainly based on the work of Liang et al. (1994). For the description of the global calibration and validation of the water balance one is referred to Nijssen et al. (2001b).

VIC version 5 (Hamman et al., 2018) upgrades did not change the model representation of physical processes, but improved the model infrastructure. Improvements include the use of NetCDF for input/output and the implementation of parallelization through Message Passing Interface (MPI). These changes increase computational speed and make VIC-5 better suited for (computationally expensive) global simulations. The most significant modification that enables new model applications is that VIC-5 also changed the processing order of the model. In previous versions all timesteps were processed for a single grid cell before continuing to the next cell (time-before-space). In VIC-5 all grid cells are processed before continuing to the next timestep (space-before-time). This development allows for interaction between grid cells every timestep, which is important for full integration of the anthropogenic impact modules, especially water withdrawals and dam operation.

For example, surface and subsurface runoff routing to produce river streamflow was typically done as a post-process operation (Lohmann et al., 1996; Hamman et al., 2017), due to the time-before-space processing order of previous versions. In order for reservoirs to account for downstream water demand, an irrigation demand initialization was required. This initialization could either be an independent offline dataset (Voisin et al., 2013a) or multiple successive model runs (Haddeland et al., 2006b). Since VIC-5 uses the space-before-time processing order, irrigation water demands and runoff routing could be simulated each timestep. The routing post-process was replaced by our newly developed routing module,

which simulates routing sequentially (upstream-to-downstream) based on the Lohmann et al. (1996)
equations.

## 2.2 Anthropogenic-impact modules

VIC-WUR extends the existing VIC-5 though the addition of several newly implemented
anthropogenic-impact modules (Fig. 1). These modules include sector-specific water withdrawal and
consumption, environmental flow requirements for both surface and groundwater systems, and dam
operation for large and small (within-grid) dams.

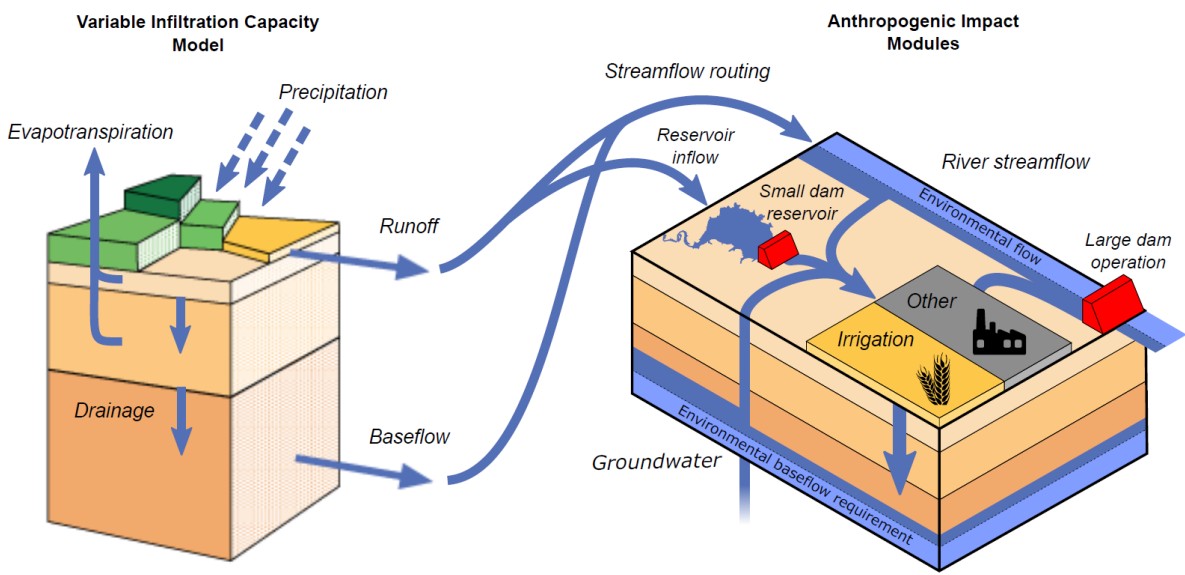


**Figure 1: Schematic overview of the VIC-WUR model that includes the VIC-5 model (left) and several anthropogenic impact modules (right). Water from river streamflow, groundwater, and small (within-grid) reservoirs are available for withdrawal. Surface and groundwater withdrawals are constrained by environmental flow requirements. Withdrawn water is available for irrigation, domestic, industrial, energy, and livestock use. Unconsumed irrigation water is returned to the soil column of the hydrological model. Unconsumed water for the other sectors is returned to the river streamflow. Small reservoirs fill using surface runoff from the cell they are located, while large dam reservoirs operate solely on rivers streamflow.**

### 2.2.1 Water withdrawal and consumption

In VIC-WUR, sectoral water demands need to be specified for each grid cell (Sect. 2.3). To meet water
demands, water can be withdrawn from river streamflow, small (within-grid) reservoirs, and
groundwater resources. Streamflow withdrawals are abstracted from the grid cell discharge (as
generated by the routing module) and reservoir withdrawals are abstracted from small dam reservoirs
(located in the cell). Groundwater withdrawals are abstracted from the third layer soil moisture and an
(unlimited) aquifer below the soil column. Aquifer abstractions represent renewable and non-renewable
abstractions from deep groundwater resources. Subsurface runoff is used to fill the aquifer if there is a
deficit.
The partitioning of water withdrawals between surface and ground water resources is data driven
(similar to e.g. Döll et al., 2012; Voisin et al., 2017; Hanasaki et al., 2018). Partitioning was based on
the study of Döll et al. (2012), who estimated groundwater withdrawal fractions for each sector in around
15.000 national and sub-national administrative units. These groundwater fractions were based mainly
on information from the International Groundwater Resources Assessment Centre (IGRAC; un-
igrac.org) database. Surface water withdrawals were partitioned between river streamflow and small
reservoirs relative to water availability. Groundwater withdrawals were first withdrawn from the third
soil layer, second from the (remaining) river streamflow resources and lastly from the groundwater
aquifer. This order was implemented to avoid overestimation of non-renewable groundwater
withdrawals as a result of errors in the partitioning data. Aquifer withdrawals are additionally limited
by the pumping capacity from Sutanudjaja et al. (2018), who estimated regional pumping capacities
based on information from IGRAC.
Water can also be withdrawn from the river streamflow of other 'remote' cells in delta areas. Since
rivers cannot split in the routing module, the model is unable to simulate the redistribution of water
resources in dendritic deltas. Therefore, streamflow at the river mouth is available for use in delta areas
(partitioned based on demand) to simulate the actual water availability. Delta areas were delineated by
the global delta map of Tessler et al. (2015).
In terms of water allocation, under conditions where water demands cannot be met, water withdrawals
are allocated to the domestic, energy, manufacturing, livestock, and irrigation sector in that order.
Withdrawn water is partly consumed, meaning the water evaporates and does not return to the
hydrological model. Consumption rates were set at 0.15 for the domestic and 0.10 for the industrial
sector, based on the data of Shiklomanov (2000). The water consumption in the energy sector was based
on Goldstein and Smith (2002) and varies per thermoelectric plant based on the fuel type and cooling
system. For the livestock sector the assumption was made that all withdrawn water is consumed.
Unconsumed water withdrawals for these sectors are returned as river streamflow. For the irrigation
sector, consumption was determined by the calculated evapotranspiration. Unconsumed irrigation water
remains in the soil column and eventually returns as subsurface runoff.

### 2.2.2    Environmental flow requirements

Water withdrawals can be constrained by environmental flow requirements (EFRs). These EFRs specify
the timing and quantity of water needed to support terrestrial river ecosystems (Smakhtin et al., 2004;
Pastor et al., 2019). Surface and groundwater withdrawals are constrained separately in VIC-WUR,
based on the EFRs for streamflow and baseflow respectively. EFRs for streamflow specify the minimum
river streamflow requirements while EFRs for baseflow specify the minimum subsurface runoff
requirements (from groundwater to surface water). Since baseflow is a function groundwater
availability, baseflow requirements are used to constrain groundwater (including aquifer) withdrawals.
Various EFR methods are available (Smakhtin et al., 2004; Richter et al., 2012; Pastor et al., 2014). Our
study used the Variable Monthly Flow (VMF) method (Pastor et al., 2014) to calculate the EFRs for
streamflows. VMF calculates the required streamflow as a fraction of the natural flow during high (30
%), intermediate (45 %) and low (60 %) flow periods, as described in Appendix B. The VMF method
performed favourably compared to other hydrological methods, in 11 case studies where EFRs were
calculated locally (Pastor et al., 2014). The advantage of the VMF method is that the method accounts
for the natural flow variability, which is essential to support freshwater ecosystems (Poff et al., 2010).
EFR methods for baseflow have been rather underdeveloped compared to EFR methods for streamflow.
However, a presumptive standard of 90 % of the natural subsurface runoff through time was proposed
by Gleeson and Richter (2018), as described in Appendix B. This standard should provide high levels
of ecological protection, especially for groundwater dependent ecosystems.
Note that part of the EFRs for baseflow are already captured in the EFRs for streamflow, especially
during low-flow periods that are usually dominated by baseflows. However, the EFRs for baseflow
specifically limit local groundwater withdrawals while EFRs for streamflow include the accumulated
runoff from upstream areas. Also, the chemical composition of groundwater derived flows is inherently
different, making them a non-substitutable water flow for environmental purposes (Gleeson and Richter,

211   2018).

### 2.2.3    Dam operation

Due to the lack of globally available information on local dam operations, several generic dam operation
schemes were developed for macro-scale hydrological models to reproduce the effect of dams on natural
streamflow (Haddeland et al., 2006a; Hanasaki et al., 2006; Zhao et al., 2016; Rougé et al., 2019; Yassin
et al., 2019). In VIC-WUR a distinction is made between 'small' dam reservoirs (with an upstream area
smaller than the cell area) and 'large' dam reservoirs, similar to Hanasaki et al. (2018), Wisser et al.
(2010a) and Döll et al. (2009). Small dam reservoirs act as buckets that fill using surface runoff of the
grid-cell they are located in and reservoirs storage can be used for water withdrawals in the same cell.
Large dam reservoirs are located in the main river and used the operation scheme of Hanasaki et al.
(2006), as described in Appendix C.
The scheme distinguishes between two dam types: (1) dams that do not account for water demands
downstream (e.g. hydropower dams or flood protection dams) and (2) dams that do account for water
demand downstream (e.g. irrigation dams). For dams that do not account for demands, dam release is
aimed at reducing annual fluctuations in discharge. For dams that do account for demands, dam release
is additionally adjusted to provide more water during periods of high demand. The operation scheme
was validated by Hanasaki et al. (2006) for 28 reservoirs and was used in various other studies (Hanasaki
et al., 2008b; Döll et al., 2009; Pokhrel et al., 2012b; Voisin et al., 2013b; Hanasaki et al., 2018). Here,
the scheme was adjusted slightly to account for monthly varying EFRs and to reduce overflow releases,
which is described in Appendix C.
The Global Reservoir and Dam (GRanD) database (Lehner et al., 2011) was used to specify location,
capacity, function (purpose), and construction year of each dam. The capacity of multiple (small- and
large) dams located in the same cell were combined.

## 2.3 Sectoral water demands

VIC-WUR water withdrawals are based on the irrigation, domestic, industry, energy, and livestock water demand in each grid-cell. Water demands represent the potential water withdrawal, which is reduced when insufficient water is available. Irrigation demands were estimated based on the hydrological model while water demands for other sectors are provided to the model as an input. Domestic and industrial were estimated based on several socioeconomic predictors, while energy and livestock water demands were derived from power plant and livestock distribution data. Due to data limitations the energy sector was incomplete, and energy water demands were partly included in the industrial water demands (which combined the remaining energy and manufacturing water demands). For more details concerning sectoral water demand calculations the reader is referred to Appendix D.

### 2.3.1 Irrigation demands

Irrigation demands were set to increase soil moisture in the root zone so that water availability is not limiting crop evapotranspiration and growth. The exception is paddy rice irrigation (Brouwer et al., 1989), where irrigation was also supplied to keep the upper soil layer saturated. Water demands for paddy irrigation practices are relatively high compared to conventional irrigation practices due to increased evaporation and percolation. Therefore, the crop irrigation demands for these two irrigation practices were calculated and applied separately (i.e. in different sub-grids). Note that multiple cropping seasons are included based on the MIRCA2000 land-use dataset (Portmann et al., 2010) (see Sect. 3.1 for more details).

Total irrigation demands also included transportation and application losses. Note that transportation and application losses are not 'lost' but rather returned to the soil column without being used by the crop. The water loss fraction was based on Frenken and Gillet (2012), who estimated the aggregated irrigation efficiency for 22 United Nations sub-regions. Irrigation efficiencies were estimated based on the difference between AQUASTAT reported irrigation water withdrawals and calculated irrigation water requirements (Allen et al., 1998), using data on crop information (e.g. growing season, harvest area) from AQUASTAT.

### 2.3.2 Domestic and industrial demands

Domestic and industrial water withdrawals were estimated based on Gross Domestic Product (GDP) per capita and Gross Value Added (GVA) by industries respectively (from Bolt et al. (2018), Feenstra et al. (2015) and World bank (2010); see Appendix D for more details). These drivers do not fully capture the multitude of socioeconomic factors that influence water demands (Babel et al., 2007). However, the wide availability of data allows for extrapolation of water demands to data-scarce regions and future scenarios (using studies such as Chateau et al. (2014)).

Domestic water demands per capita (used for drinking, sanitation, hygiene, and amenity uses) were estimated similar to Alcamo et al. (2003). Demands increased non-linearly with GDP per capita due to the acquisition of water using appliances as household become richer. A minimum water supply is needed for survival, and the saturation of water using appliances sets a maximum on domestic water demands. Industrial water demands (used for cooling, transportation, and manufacturing) were estimated similar to Flörke et al. (2013) and Voß and Flörke (2010). Industrial demands increased linearly with GVA (as an indicator of industrial production). Since industrial water intensities (i.e. the water use per production unit) vary widely between different industries (Flörke and Alcamo, 2004 ; Vassolo and Döll, 2005; Voß and Flörke, 2010), the average water intensity was estimated for each country. Both domestic and industrial water demands were also influenced by technological developments that increase water-use efficiency over time, as in Flörke et al. (2013).

Domestic water demands varied monthly based on air temperature variability as in Huang et al. (2018) (based on Wada et al. (2011b)). Using this approach, water demands were higher in summer than in winter, especially for counties with strong seasonal temperature differences. Domestic water demand per capita were downscaled using the HYDE3.2 gridded population maps (Goldewijk et al., 2017). Industrial water demands were kept constant throughout the year. Industrial demands were downscaled from national to grid cell values using the NASA Back Marble night-time light intensity map (Roman et al., 2018). National industrial water demands were allocated based on the relative light intensity per grid cell for each country.

### 2.3.3  Energy and livestock demands

Energy water demands (used for cooling of thermoelectric plants) were estimated using data from van Vliet et al. (2016). Water use intensity for generation (i.e. the water use per generation unit) was estimated based on the fuel and cooling system type (Goldstein and Smith, 2002), which was combined with the generation capacity. Note that the data only covered a selection of the total number of thermoelectric power plants worldwide. Around 27 % of the total (non-renewable) global installed capacity between 1980 and 2011 was included in the dataset due to lack of information on cooling system types for the majority of thermoelectric plants. To avoid double counting, energy water demands were subtracted from the industrial water demands.

Livestock water demands (used for drinking and animal servicing) were estimated by combining the Gridded Livestock of the World (GLW3) map (Gilbert et al., 2018) with the livestock water requirement reported by Steinfeld et al. (2006). Eight varieties of livestock were considered: cattle, buffaloes, horses, sheep, goats, pigs, chicken, and ducks. Drinking water demands varied monthly based on temperature as described by Steinfeld et al. (2006), whereby drinking water requirements were higher during higher temperatures.

## 3  Model application

### 3.1  Setup

VIC-WUR results were generated between 1979 and 2016, excluding a spin-up period of one year (analysis period from 1980 to 2016). The model used a daily timestep (with a 6-hourly timestep for snow processes) and simulations were executed on a 0.5° by 0.5° grid (around 55 km at the equator) with three soil layers per grid cell. Soil and (natural) vegetation parameters were the same as in Nijssen et al. (2001c) (disaggregated to 0.5°), who used various sources to determine the soil (Cosby et al., 1984; Carter and Scholes, 1999) and vegetation parameters (Calder, 1993; Ducoudre et al., 1993; Sellers et al., 1994; Myneni et al., 1997).

Nijssen et al. (2001c) used the Advanced Very High Resolution Radiometer vegetation type database (Hansen et al., 2000) to spatially distinguish 13 land cover types. The land cover type 'cropland' in the

original land-cover dataset was replaced by cropland extents from the MIRCA2000 cropland dataset
(Portmann et al., 2010). MIRCA2000 distinguishes the monthly growing area(s) and season(s) of 26
irrigated and rain-fed crop types around the year 2000. Crop types were aggregated into three land cover
types: rain-fed, irrigated, and paddy rice cropland. The natural vegetation was proportionally rescaled
to make up discrepancies between the natural vegetation and cropland extents.
Cropland coverage (the cropland area actually growing crops) varied monthly based on the crop growing
areas of MIRCA2000. The remainder was treated as bare soil. Cropland vegetation parameters (e.g. Leaf
Area Index (LAI), displacement, vegetation roughness and albedo) vary monthly based on the crop
growing seasons and the development-stage crop coefficients of the Food and Agricultural Organisation
(Allen et al., 1998).
The latest WATCH forcing data Era Interim (aggregated to 6 hourly), developed by the EU Water and
Global Change (WATCH; Harding et al., 2011) project, was used as climate forcing (WFDEI; Weedon
et al., 2014). The dataset provides gridded historical climatic variables of minimum and maximum air
temperature, precipitation (as the sum of snowfall and rainfall, GPCC bias-corrected), relative humidity,
pressure, and incoming shortwave and longwave radiation.
For naturalized simulations only the routing module was used. For the human-impact simulations the
sectoral water withdrawals and dam operation modules were turned on in the model simulations. For
the EFR-limited simulations water withdrawals and dam operations were constrained as described.
**3.2    Validation and evaluation**
In order to validate the VIC-WUR human-impact modules, water withdrawal, terrestrial total water
storage anomalies, and streamflow and reservoir operation simulations were compared with
observations. The validation specifically focused on the effects of the newly included human-impact
modules, meaning that streamflow and total-water storage anomaly results are shown for river basins
that are strongly influenced by human activities. A general validation for streamflow and terrestrial total
water storage anomalies (including basins with limited human activities) is shown in Appendix E.

### 3.2.1 Sectoral water withdrawals

Simulated global domestic, industrial, livestock, and irrigation mean water withdrawals were 310, 771, 36, and 2202 km$^3$ year$^{-1}$ respectively for the period of 1980 to 2016. Sectoral water withdrawals were compared with FAO national annual water withdrawals (FAO, 2016), monthly withdrawal data from Huang et al. (2018), and annual withdrawal data from Shiklomanov (2000) and Steinfeld et al. (2006). For the latter studies, water withdrawals were aggregated by region (world, Africa, Asia, Americas, Europe and Oceania). Note that Huang et al. (2018) irrigation water withdrawals integrate results of four other macro-scale hydrological models (WaterGAP, H08, LPJmL, PCR-GLOBWB), using the same land-use and climate setup as our study. Results from individual macro-scale hydrological models are also shown.

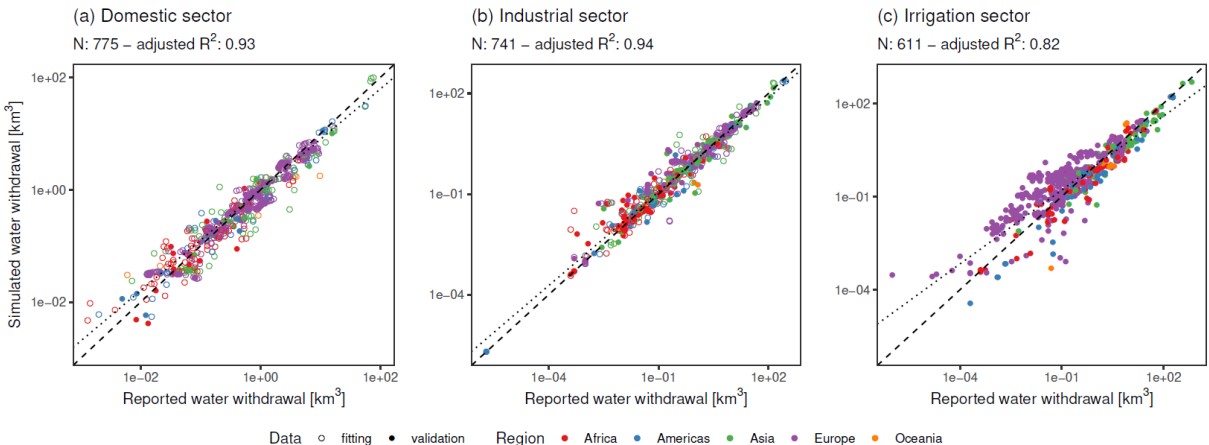

**Figure 2: Comparison between simulated and FAO reported national annual water withdrawals for the (a) domestic, (b) industrial, and (c) irrigation sector. Colours distinguish between regions. Open circles were also used in the calibration of the water withdrawal demands. The dashed line indicates the 1:1 ratio and the spotted line indicates the simulated best linear fit. Note the log-log axis which is used to display the wide range of water withdrawals. The adjusted R$^2$ is also based on the log values.**

Simulated domestic, industrial, and irrigation water withdrawals correlated well to reported national water withdrawals, with adjusted R$^2$ of 0.93, 0.94, and 0.82 for domestic, industrial, and irrigation water withdrawal respectively (Fig. 2a-c). Generally, smaller water withdrawals were overestimated and larger water withdrawals were underestimated. Differences for the domestic and industrial sector were small and probably related to the fact that smaller countries were poorly delineated on a 0.5° by 0.5° grid. However, irrigation differences were larger with overestimations of irrigation water withdrawals in (mostly) Europe. Since irrigation water demands are the results of the simulated water balance,

overestimations would indicate a regional underestimation of water availability for Europe or differences in irrigation efficiency.

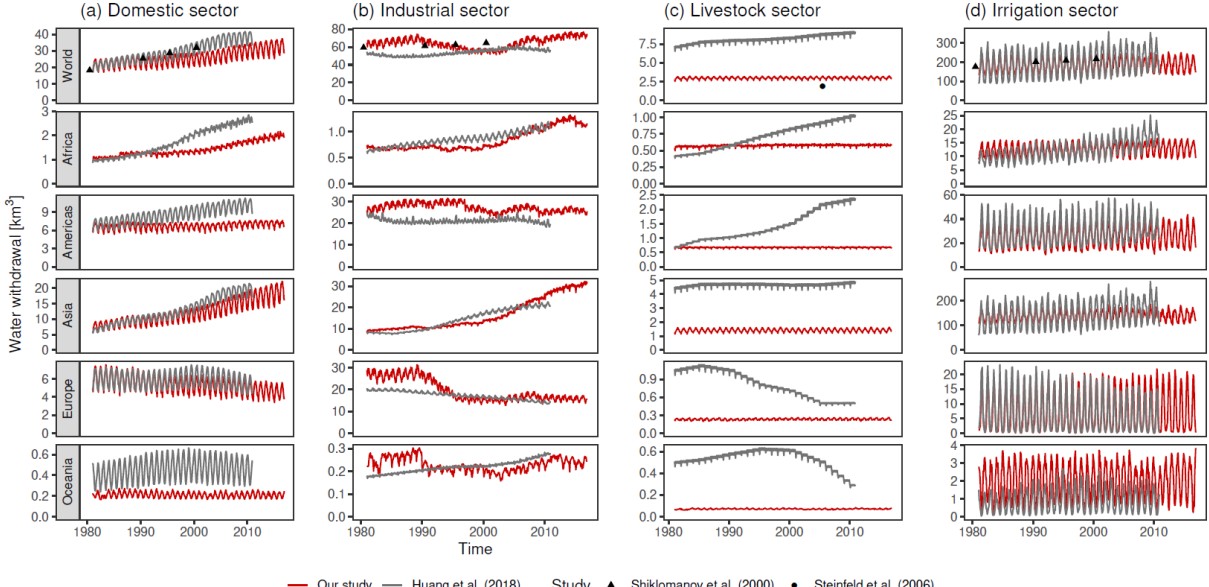

**Figure 3: Comparison between simulated and Huang et al. (2018), Shiklomanov (2000), and Steinfeld et al. (2006) compiled monthly and annual regional water withdrawals for the (a) domestic sector, (b) industrial sector, (c) livestock sector, and (d) irrigation. Colours and shapes distinguish between studies. Note that the jitter in livestock withdrawals is due to the different days per month.**

When domestic, industrial, and livestock water withdrawals were compared to other studies, results were mixed (Fig. 3a-c). Simulated domestic withdrawals followed a similar trend in time. However, simulated domestic water withdrawals trends were overall somewhat underestimated with a mean bias of 54 km$^3$ year$^{-1}$ compared to Huang et al. (2018). Asia is the main contributor to the global underestimation, but results are similar in most regions. Simulated industrial water withdrawal were (mostly) higher in our study with a mean bias of 107 km$^3$ year$^{-1}$ compared to Huang et al. (2018) but only a mean bias of 5 km$^3$ year$^{-1}$ compared to Shiklomanov (2000). Also, industrial water withdrawal trends in time were less consistent.

Withdrawal differences for the domestic and industrial sector are probably due to the limited data availability. Our approach to compute water demands was data-driven and sensitive to data gaps (as opposed to Huang et al. (2018) who also combined model results). For example, domestic withdrawal data for China was not available before 2007 and industrial withdrawal data was limited before 1990.

Also, data on the disaggregation of industrial sectors (e.g. energy and mining) was limited, which can
be important sectors in the water-food-energy nexus.
For livestock water withdrawals there is a large discrepancy between the Huang et al. (2018) and
Steinfeld et al. (2006). Both studies used similar livestock maps, but there was large differences in
livestock water intensity [litre animal$^{-1}$ year$^{-1}$]. Since our study used Steinfeld et al. (2006) to estimate
livestock water intensity, our results were closer to their values (slightly higher due to the inclusion of
buffaloes, horses, and ducks). Note that Huang et al. (2018) shows trends in livestock water withdrawals
while our study used static livestock maps.
**Table 1: Average annual global irrigation water withdrawals as calculated by several global hydrological models.**
**\*Includes livestock withdrawals.**

| Model | Irrigation withdrawal [km3 year-1] | Representative years | Reference |
|---|---|---|---|
| VIC-WUR | 2202 (± 60) | 1980-2016 | Our study |
| H08 | (a) 2810<br>(b) 2544 (± 75) | (a) 1995<br>(b) 1984 - 2013 | (a) Hanasaki et al. (2008a)<br>(b) Hanasaki et al. (2018) |
| MATSIRO | (a) 2158 (± 134)<br>(b) 3028 (± 171) | (a) 1983 - 2007<br>(b) 1998 - 2002 | (a) Pokhrel et al. (2012a)<br>(b) Pokhrel et al. (2015) |
| LPJmL | 2555 | 1971 - 2000 | Rost et al. (2008) |
| PCR-GLOB | (a) 2644<br>(b) 2309 * | (a) 2010<br>(b) 2000 - 2015 | (a) Wada and Bierkens (2014)<br>(b) Sutanudjaja et al. (2018) |
| WaterGAP | (a) 3185<br>(b) 2400 | (a) 1998-2002<br>(b) 2003 - 2009 | (a) Döll et al. (2012)<br>(b) Döll et al. (2014) |
| WBM | 2997 | 2002 | Wisser et al. (2010b) |

Simulated irrigation water withdrawals were within range of other macro-scale hydrological model
estimates (Table 1). Simulated monthly variability in irrigation water withdrawals is reduced compared
to the compiled results of Huang et al. (2018) (Fig. 3d), especially in Asia. Also, trends in time are less
pronounces as can be seen in Africa. These differences may indicate a relative low weather/climate
sensitivity of evapotranspiration in VIC-WUR, as annual and interannual weather changes affect
irrigation water demands to a lesser degree.

### 3.2.2 Groundwater withdrawals and depletion

Simulated global mean withdrawals were 2327 and 992 km$^3$ year$^{-1}$ for surface and groundwater respectively for the period of 1980 to 2016. Of the global groundwater withdrawals, 334 km$^3$ year$^{-1}$ contributed to groundwater depletion. Simulated ground and surface water withdrawals and terrestrial total water storage anomalies were compared FAO national annual water withdrawals (FAO, 2016) and monthly storage anomaly data from the GRACE satellite (NASA, 2002). GRACE satellite total water storage anomalies were used to validate total water storage dynamics as well as groundwater exploitation contributing to downward trends in total water storage. Groundwater depletion results from other macro-scale hydrological models are shown as well. In order to compare the simulation results to the GRACE dataset, a 300km gaussian filer was applied to the simulated data (similar to Long et al. (2015)).

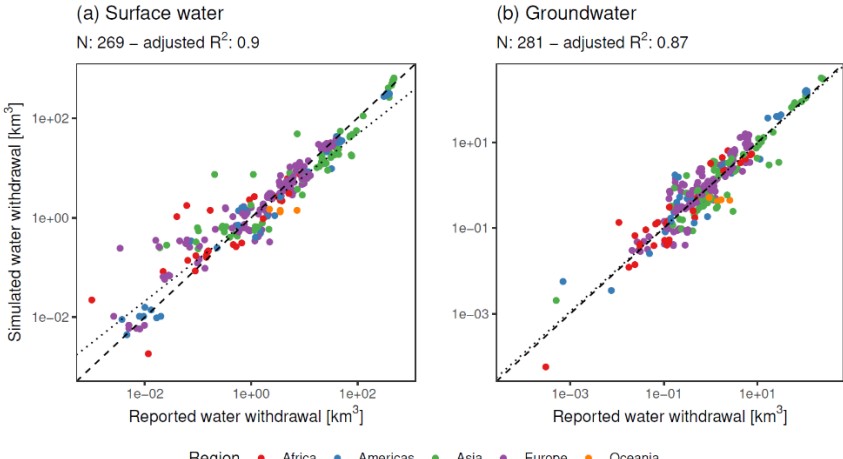

**Figure 4: Comparison between simulated and FAO reported national annual water withdrawals from (a) surface water and (b) groundwater. Colours distinguish between regions. The dashed line indicates the 1:1 ratio and the spotted line indicates the simulated best linear fit. Note the log-log axis which is used to display the wide range of water withdrawals. The adjusted R$^2$ is also based on the log values.**

Simulated surface and groundwater withdrawals correlated well to the reported national water withdrawals, with adjusted R$^2$ of 0.90 and 0.87 for surface and groundwater respectively (Fig. 4a-b). Surface water withdrawals were overestimated for low withdrawals and underestimated for large withdrawals. There is a weak correlation (-0.35) between the underestimations in surface water withdrawals and the overestimation in groundwater withdrawals, meaning water withdrawal differences could be related to the partitioning between surface and groundwater resources. Also, it is likely that

low water demands are overestimated (as discussed in Sect. 3.2.1), resulting in an overestimation of low

surface water withdrawals.

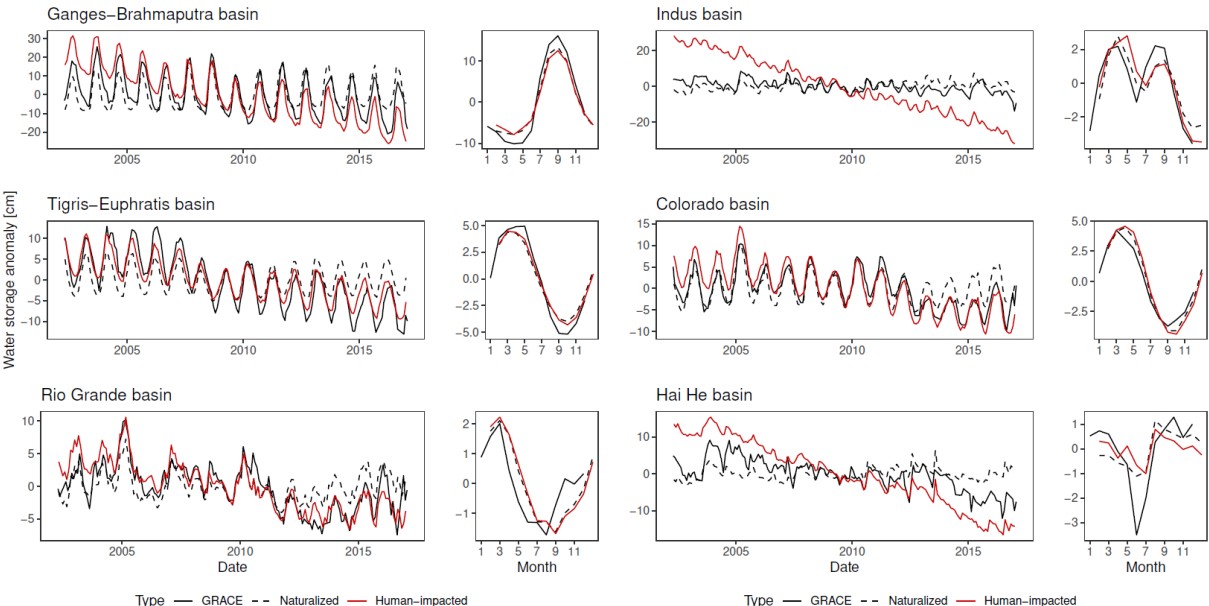

**Figure 5: Comparison between simulated and GRACE observed monthly terrestrial total water storage anomalies. Figures indicate timeseries and multi-year mean average for naturalized simulations (dashed), human-impacted simulations (red), and observed (black) terrestrial total water storage anomalies.**

Simulated monthly terrestrial water storage anomalies correlated well to the GRACE observations, with

mean annual and inter-annual Root Mean Squared Error (RMSE) of 1.9 mm and 3.5 mm respectively.

The difference between annual and inter annual performance was primarily due to the groundwater

depletion process (Fig. 5). Simulated groundwater depletion was (mostly) overestimated (e.g. Indus and

Hai He basins), with higher declining trends in terrestrial total water storage for most basins. However,

compared to other macro-scale hydrological models, simulated groundwater withdrawal and

exploitation was within range (Table 2), even though total groundwater withdrawals were relatively

high.

**Table 2: Average annual global groundwater withdrawals and depletion as calculated by several global hydrological models.**

| Model | Groundwater withdrawal [km3 year-1] | Groundwater depletion [km3 year-1] | Representative years | Reference |
|---|---|---|---|---|
| VIC-WUR | 992 (± 51) | 316 (± 63) | 1980 - 2016 | Our study |
| H08 | 789 (± 30) | 182 (± 26) | 1984 - 2013 | Hanasaki et al. (2018) |

| | | | |
|---|---|---|---|
| MATSIRO | 570 (± 61) | 330 | 1998 - 2002 | Pokhrel et al. (2015) |
| GCAM | | (a) 600 | (a) 2005 | (a) Kim et al. (2016) |
| | | (b) 550 | (b) 2000 | (b) Turner et al. (2019) |
| PCR-GLOB | (a) 952 | (a) 304 | (a) 2010 | (a) Wada and Bierkens (2014) |
| | (b) 632 | (b) 171 | (b) 2000 - 2015 | (b) Sutanudjaja et al. (2018) |
| WaterGAP | (a) 1519 | (a) 250 | (a) 1998-2002 | (a) Döll et al. (2012) |
| | (b) 888 | (b) 113 | (b) 2000 - 2009 | (b) Döll et al. (2014) |

As with the FAO comparison, these results seems to indicate that withdrawal partitioning towards
groundwater is overestimated. However, conclusions regarding groundwater depletion are limited by
the relatively simplistic approach to groundwater used in our study (as discussed by Konikow (2011)
and de Graaf et al. (2017)). For example, processes such as wetland recharge and groundwater flows
between cells are not simulated, even though these could decrease groundwater depletion.

### 3.2.3    Discharge modification

Simulated discharge was compared to GRDC station data (GRDC, 2003) for various human-impacted
rivers. Stations were selected if the upstream area was larger than 20,000 km2, matched the simulated
upstream area at the station location, and the available data spanned more than 2 years. Subsequently,
stations where the human-impact modules did not sufficiently impacted discharge were omitted. In order
validate the reservoir operation more thoroughly, simulated reservoir inflow, storage, and release was
compared with operation data from Hanasaki et al. (2006) and Yassin et al. (2019). Reservoirs were
included if the simulated storage capacity (which is the combined storage capacity of all large dams in
a grid) was similar to observed storage capacity.

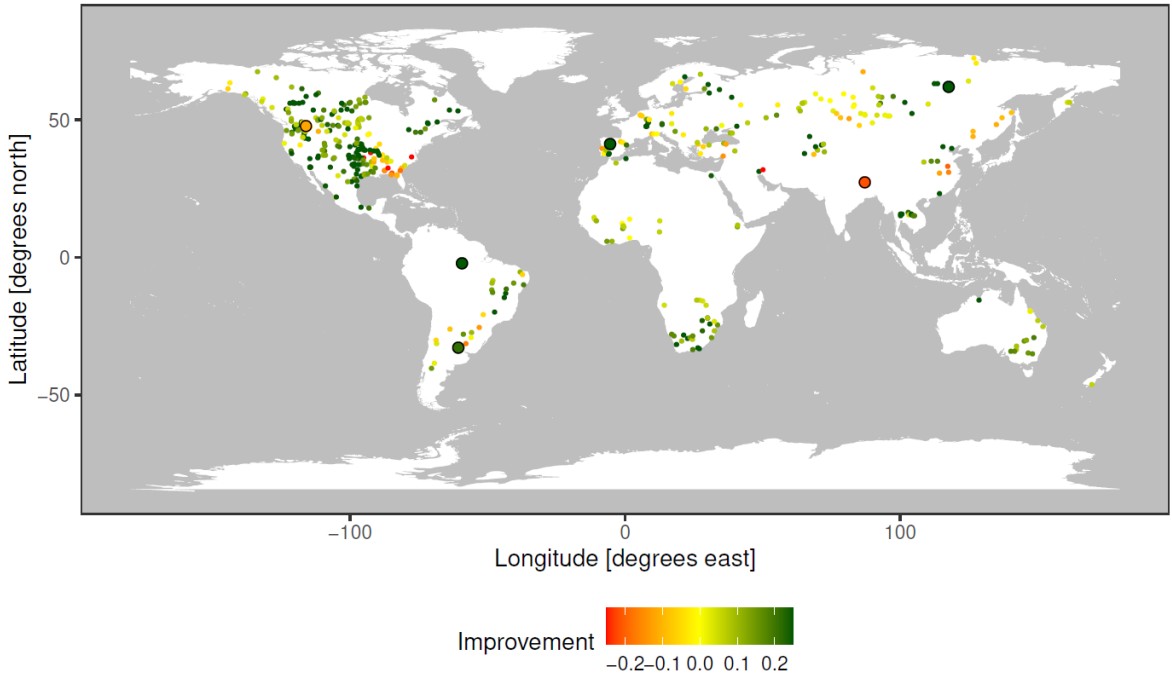


**Figure 6: Discharge improvement from naturalized to human-impacted simulations (as a fraction of the naturalized RMSE). Circled larger stations are shown in Fig. 7.**

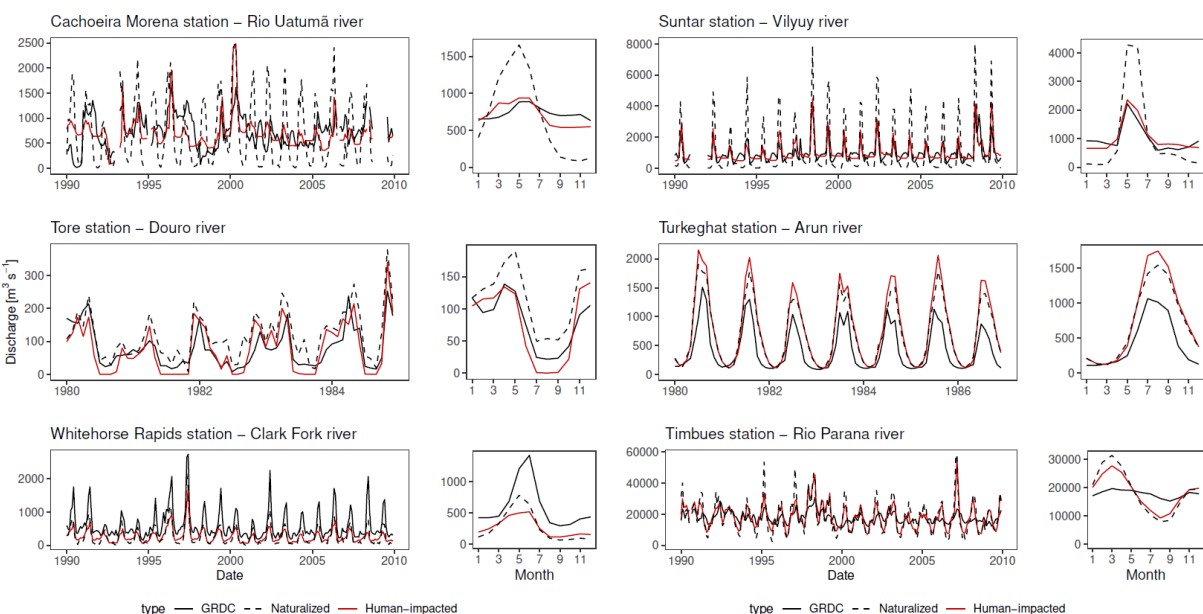


**Figure 7: Comparison between simulated and GRDC observed discharge. Figures indicate timeseries and multi-year average of for naturalized simulations (dashed), human-impacted simulations (red), and observed (black) discharge.**

The inclusion of the human-impact modules improved discharge performance, measured in RMSE, for 370 out of 462 stations (80 %; Fig. 6 and 7). Improvements were mainly due to the effects of reservoir operation on discharges (e.g. Cachoeira Morena and Suntar stations), but also due to withdrawal

reductions (e.g. Tore station). Reservoir effects on discharge were sometimes underestimated however
(e.g. Timbues station).
Decreased performance was mostly related to under or overestimations of (calibrated) natural
streamflow which was subsequently exacerbated by reservoir operation and water withdrawals. For
example, the Clark Fork river naturalized streamflow was underestimated, which was subsequently
further underestimated by the human-impact modules (Whitehorse Rapids station). Also, increases in
discharge due to groundwater withdrawals could increase naturalized streamflow (e.g. Turkeghat
station). Further improvements to discharge performance would most likely require either a recalibration
of the VIC model parameters.

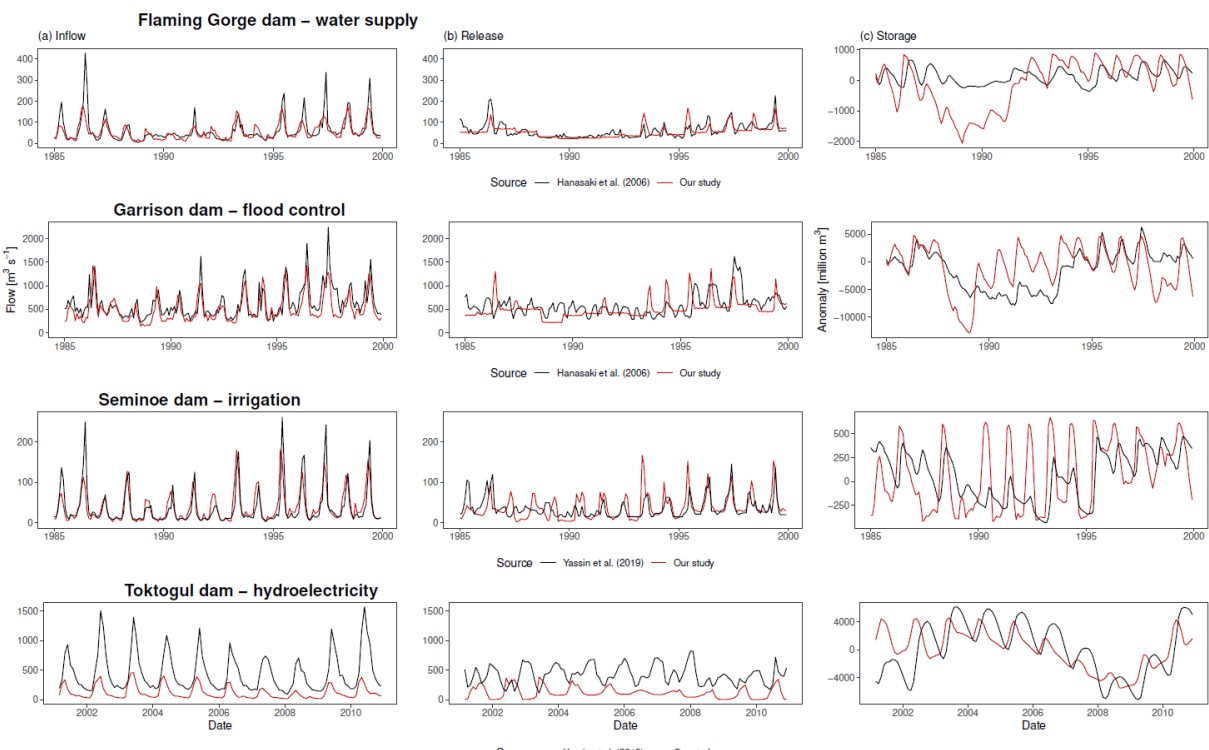


**Figure 8: Comparison between simulated and Hanasaki et al. (2006) and Yassin et al. (2019) observed reservoir operation. Figures indicate timeseries and multi-year averages of (a) inflow, (b) release, and (c) storage anomalies for human-impacted simulations (red) and observations (black).**

For individual reservoirs, operation characteristics were generally well simulated (Fig. 8), with
reductions in annual discharge variations (e.g. Flaming Gorge and Garrison dams) and increased water
release for irrigation (e.g. Seminoe dam). However, due to changes in locally simulated and actual
inflow, dam operation can take on different characteristics (e.g. Toktogul dam). Also, peak discharge
events caused by reservoir overflow (as also described by Masaki et al. (2018)) were not always
sufficiently represented in the observations (e.g. Garisson dam). These differences indicate locally
varying reservoir operation strategies. Several studies have developed reservoir operation schemes that
can be calibrated to the local situation (Rougé et al., 2019; Yassin et al., 2019). However, worldwide
implementations of these operation schemes remains limited by data availability.
**3.3    Integrated environmental flow requirements**
In order to assess the impact and capabilities of the newly integrated environmental flow requirements
(EFRs) module, simulated water withdrawals with and without adhering to EFRs were compared.

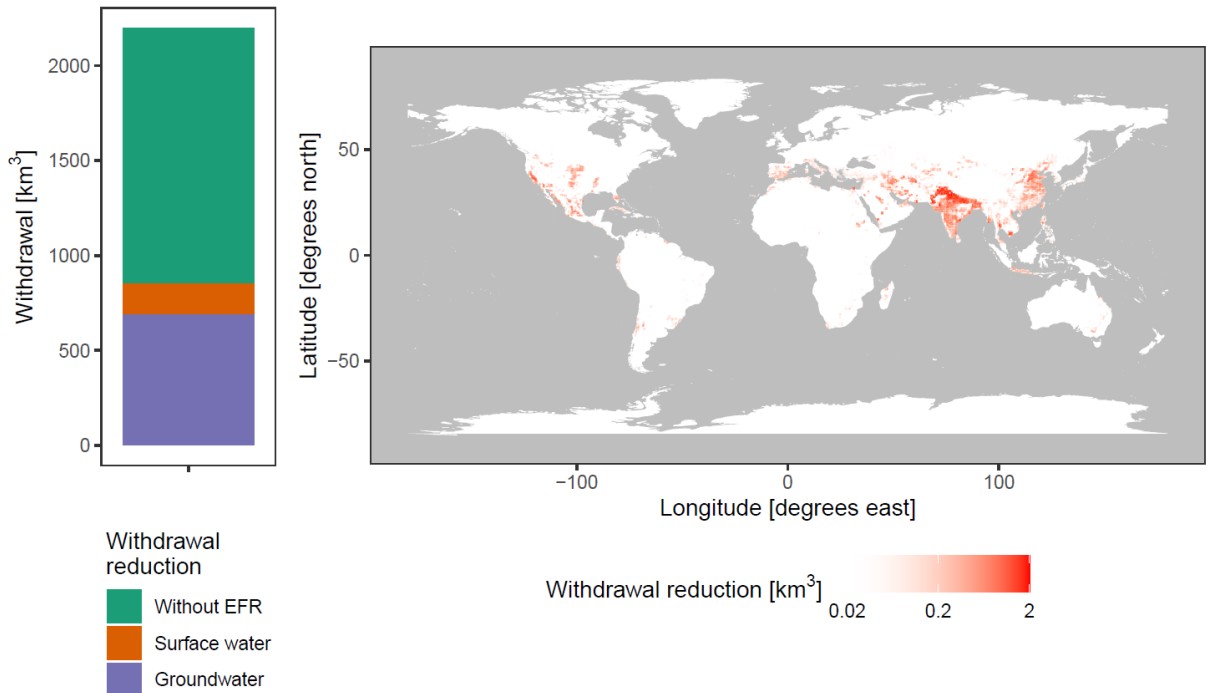


**Figure 9: Average annual irrigation water withdrawal reductions when adhering to EFRs as global gross total (left) and**
**spatially distributed (right). Global gross totals are separated into withdrawals without any reduction (green), surface**
**water withdrawal reductions (orange), and groundwater withdrawal reductions (purple). Note the log axis for the**
**spatially distributed withdrawal reductions to better display the spatial distribution of the reductions.**
If water-use would be limited to EFRs, irrigation withdrawals would need to be reduced by about 39 %
(851 km$^3$ year$^{-1}$) (Fig. 9a). Under the strict requirements used in our study, 81 % (693 km$^3$ year$^{-1}$) of the
reduction could be attributed to limitations imposed on groundwater withdrawals. Subsequently, the
impact of the environmental flow requirements (if adhered to) would be largest in groundwater
dependent regions (Fig. 9b). Note that, due to the full integration of EFRs, downstream surface water
withdrawals increased by 98 $km^3$ $year^{-1}$ when limiting groundwater withdrawals on top of limiting
surface water withdrawals, due to increase subsurface runoff.
Reductions due to EFRs were similar to Jägermeyr et al. (2017), who calculated irrigation withdrawal
reductions of 41 % (997 $km^3$ $year^{-1}$) assuming only surface water abstractions. In our study, surface
water reductions were smaller since the strict groundwater requirements increases subsurface runoff to
surface waters. It can be discussed to what extent the EFRs for baseflow were too constricting, since
they were based on the relatively stringent EFR for streamflow of Richter et al. (2012) (10 % of the
natural streamflow). However, in the absence of any other standards, this baseflow standard remains the
best available. Note that, even when accounting for EFRs for baseflow on a grid scale, withdrawals
could still have local and long-term impacts that are not captured by the model. The timing, location,
and depth of groundwater withdrawals are important factors due to their interactions with the local
geohydrology, as discussed by Gleeson and Richter (2018).
**4    Conclusions**
The VIC-WUR model introduced in this paper aims to provide new opportunities for global water
resource assessments using the VIC model. Accordingly, several anthropogenic impact modules, based
on previous major works, were integrated into the VIC-5 macro-scale hydrological model: domestic,
industrial, energy, livestock, and irrigation water withdrawals from both surface water and groundwater
as well as an integrated environmental flow requirement module and dam operation module. Global
gridded datasets on domestic, industrial, energy, and livestock demand were developed separately and
used to force the VIC-WUR model.
Simulated national water withdrawals were in line with reported national annual withdrawals (adjusted
$R^2 > 0.8$; both per sector as per source). However, the data-oriented methodology used to derive sectoral
water demands resulted in different withdrawal trends over time compared to other studies
(Shiklomanov, 2000; Huang et al., 2018). While the current setup to estimate sectoral water demands is
well suited for future water withdrawal estimations, there are various other approaches (e.g. Alcamo et
al., 2003; Vassolo and Döll, 2005; Shen et al., 2008; Hanasaki et al., 2013; Wada and Bierkens, 2014).
As the model setup of VIC-WUR allows for the evaluation of other sectoral water demand inputs (on
various temporal aggregations), several different approaches can be used depending on the focus region
and data-availability for calibration. Terrestrial water storage anomaly trends were well simulated (mean
annual and inter-annual RMSE of 1.9 mm and 3.6 mm respectively), while groundwater exploitation
was overestimated. Overestimated groundwater depletion rates are likely related to an over-partitioning
of water withdrawals to groundwater. The implemented human impact modules increased simulated
discharge performance (370 out of 462 stations), mostly due to the effects of reservoir operation.
An assessment of the effect of EFRs shows that, when one would adhere to these requirements, global
water withdrawals would be severely limited (39 %). This limitation is especially the case for
groundwater withdrawals, which, under the strict requirements used in our study, need to be reduced by

526   81 %.

VIC-WUR has potential for studying impacts of climate change and anthropogenic developments on
current and future water resources and sectoral specific-water scarcity. The additions presented here
make the VIC model more suited for fully-integrated worldwide water-resource assessments and
substantially decrease computation times compared to Haddeland et al. (2006a).
**5   Appendices**
**5.1   Appendix A: VIC water and energy balance**
In VIC each sub-grid computes the water and energy balance individually (i.e. sub-grid do not exchange
water or energy between one another). For the water balance, incoming precipitation is partitioned
between evapotranspiration, surface and subsurface runoff, and soil water storage. Potential
evapotranspiration is based on the Penman-Monteith equation without the canopy resistance
(Shuttleworth, 1993). The actual evapotranspiration is calculated by two methods, based on whether the
land cover is vegetated or not (bare soil). Evapotranspiration of vegetation is constrained by stomatal,
architectural and aerodynamic resistances and is partitioned between canopy evaporation and
transpiration based on the intercepted water content of the canopy (Deardorff, 1978; Ducoudre et al.,
1993). Bare soil evaporation is constrained by the saturated area of the upper soil layer. The saturated

area is variable within the grid since (as the model name implies) the infiltration capacity of the soil is assumed heterogeneous (Franchini and Pacciani, 1991). Saturated areas evaporate at the potential evaporation rate while in unsaturated areas evaporation is limited. Surface runoff is produced by precipitation over saturated areas. Precipitation over unsaturated areas infiltrates into the upper soil layer and drains through the soil layers based on the gravitational hydraulic conductivity equations of Brooks and Corey (1964). In the first and second layer water is available for transpiration, while the third layer is assumed to be below the root zone. From the third layer baseflow is generated based on the non-linear Arno conceptualization (Franchini and Pacciani, 1991). Baseflow increases linearly with soil moisture content when the moisture content is low. At higher soil moisture contents the relation is non-linear, representing subsurface storm-flows.

For the energy balance, incoming net radiation is partitioned between sensible, latent, and ground heat fluxes and energy storage in the air below the canopy. The energy storage below the canopy is omitted if it is considered negligible (e.g. the canopy surface is open or close to the ground). The latent heat flux is determined by the evapotranspiration as calculated in the water balance. The sensible heat flux is calculated based on the difference between the air and surface temperature and the ground heat flux is calculated based on the difference between the soil and surface temperature. Since the incoming net radiation is also a function of the surface temperature (specifically the outgoing longwave radiation), the surface temperature is solved iteratively. Subsurface ground heat fluxes are calculated assuming an exponential temperature profile between the surface and the bottom of the soil column, where the bottom temperature is assumed constant. Later model developments included options for finite difference solutions of the ground temperature profile (Cherkauer and Lettenmaier, 1999), spatial distribution of soil temperatures (Cherkauer and Lettenmaier, 2003), a quasi-2-layer snow-pack snow model (Andreadis et al., 2009), and blowing snow sublimation (Bowling et al., 2004).

## 5.2 Appendix B: EFRs for surface and groundwater

VIC-WUR used the Variable Monthly Flow (VMF) method (Pastor et al., 2014) to limit surface water withdrawals. The VMF method (Pastor et al., 2014) calculates the EFRs for streamflow as a fraction of the natural flow during high (Eq. A.1), intermediate (Eq. A.2), and low (Eq. A.3) flow periods. The

presumptive standard Gleeson and Richter (2018) is used to limit groundwater withdrawals (including
aquifer groundwater withdrawals). This standard calculates the EFRs for baseflow as 90 % of the natural
subsurface runoff through time (Eq. A.4). Here, daily instead of monthly EFRs were used to better
capture the monthly flow variability.
$EFR_{s,d} = 0.6 \cdot NF_{s,d}$                                                                    Eq. (A.1)
$where\ NF_{s,d} \leq 0.4 \cdot NF_{s,y}$
$EFR_{s,d} = 0.45 \cdot NF_{s,d}$                                                                  Eq. (A.2)
$where\ 0.4 \cdot MF_{s,y} < NF_{s,d} \leq 0.8 \cdot NF_{s,y}$
$EFR_{s,d} = 0.3 \cdot NF_{s,d}$                                                                    Eq. (A.3)
$where\ NF_{s,d} > 0.8 \cdot NF_{s,y}$
$EFR_{b,d} = 0.9 \cdot NF_{b,d}$                                                                    Eq. (A.4)
Where $EFR_{s,d}$ is the daily EFRs for streamflow [m$^3$ s$^{-1}$], $EFR_{b,d}$ the daily EFRs for baseflow [m$^3$ s$^{-1}$],
$NF_{s,d}$ is the average natural daily streamflow [m$^3$ s$^{-1}$], and $NF_{s,y}$ is the average natural yearly streamflow
[m$^3$ s$^{-1}$], and $NF_{b,d}$ is the average natural daily baseflow [m$^3$ s$^{-1}$].
EFRs for streamflow and baseflow were based on VIC-WUR naturalized simulations between 1980 and
2010. Average natural daily flows were calculated as the interpolated multi-year monthly average flow
over the simulation period.
**5.3   Appendix C: Dam operation scheme**
VIC-WUR used a dam operation scheme based on Hanasaki et al. (2006). Target release (i.e. the
estimated optimal release) was calculated at the start of the operational year. The operational year starts
at the month where the inflow drops below the average annual inflow, and thus the storage should be at
its desired maximum. The scheme distinguished between two dam types: (1) dams that did not account
for water demands downstream (e.g. hydropower dams or flood control) and (2) dams that did account
for water demands downstream (e.g. irrigation dams). The original scheme of Hanasaki et al. (2006)
also accounts for EFRs, which were fixed at half the annual mean inflow. Other studies lowered the
requirements to a tenth of the mean annual inflow, increasing irrigation availability and preventing
excessive releases (Biemans et al., 2011; Voisin et al., 2013b). In our study the original dam operation
scheme was adapted slightly to account for monthly varying EFRs.
For dams that did not account for demands, the initial release was set at the mean annual inflow corrected
by the variable EFRs (Eq. A.5). For dams that did account for demands, the initial release was increased
during periods of higher water demand. If demands were relatively high compared to the annual inflow,
the release was corrected by the demand relative to the mean demand (Eq. A.6). If demands were
relatively low compared to the annual inflow, release was corrected based on the actual water demand
(Eq. A.7).

$R'_m = EFR_{s,m} + (I_y - EFR_{s,y})$                           Eq. (A.5)

$where\ D_y = 0$

$R'_m = EFR_{s,m} + \left(I_y - EFR_{s,y}\right) * \frac{D_m}{D_y}$                Eq. (A.6)

$where\ D_y > 0\ and\ D_y > (I_y - EFR_{s,y})$

$R'_m = EFR_{s,m} + (I_y - EFR_{s,y}) - D_y + D_m$          Eq. (A.7)

$where\ D_y > 0\ and\ D_y \le (I_y - EFR_{s,y})$

Where $R'_m$ is the initial monthly target release [m$^3$ s$^{-1}$], $EFR_{s,m}$ is the average monthly EFR for
streamflow demand [m$^3$ s$^{-1}$], $I_y$ is the average yearly inflow [m$^3$ s$^{-1}$], $EFR_{s,y}$ is the average yearly EFR
for streamflow [m$^3$ s$^{-1}$], $D_m$ is the average monthly water demand [m$^3$ s$^{-1}$], and $D_y$ is the average yearly
water demand [m$^3$ s$^{-1}$].
As in Hanasaki et al. (2006), the initial target release was adjusted based on storage and capacity. Target
release was adjusted to compensate differences between the current storage and the desired maximum
storage (Eq. A.8). Target release was additionally adjusted if the storage capacity is relatively low
compared to the annual inflow, and unable to store large portions of the inflow for later release (Eq.
A.9).
$R_m = k \cdot R'_m$ Eq. (A.8)
*where* $c \geq 0.5$
$R_m = \left(\frac{c}{0.5}\right)^2 \cdot k \cdot R'_m + \left\{1 - \left(\frac{c}{0.5}\right)^2\right\} \cdot I_m$ Eq. (A.9)
*where* $0 \leq c \leq 0.5$
Where $I_m$ is the average monthly inflow [m$^3$ s$^{-1}$], $c$ the capacity parameter [-] calculated as the storage
capacity divided by the mean annual inflow, and $k$ the storage parameter [-] calculated as current storage
divided by the desired maximum storage. The desired maximum storage was set at 85 % of the storage
capacity as recommended by Hanasaki et al. (2006).
Water inflow, demand and EFRs were estimated based on the average of the past five years. Water
demands were based on the water demands of downstream cells. Only a fraction of water demands were
taken into account, based on the fraction of discharge the dam controlled. For example: if a dam
controlled 70 % of the discharge of a downstream cell, than 70 % of its demands were taken into account.
Fractions smaller than 25 % were ignored.
The original dam operation scheme of Hanasaki et al. (2006) was shown to produce excessively high
discharge events due to overflow releases (Masaki et al., 2018). These overflow releases occurred due
to a mismatch between the expected and actual inflow. In our study, dam release was increased during
high-storage events to prevent overflow and accompanying high discharge events. If dam storage was
above the desired maximum storage, target dam release was increased to negate the difference (Eq.
A.10). If dam storage was below the desired minimum storage, release is decreased (Eq. A.11). Dam
release was adjusted exponentially based on the relative storage difference: small storage differences
were only corrected slightly, but if the dam was close to overflowing or emptying, the difference was
corrected strongly.
$$R_a = R_m + \frac{(S-C\alpha)}{\gamma} \cdot \left(\frac{\frac{S}{C}-\alpha}{1-\alpha}\right)^b$$ Eq. (A.10)
$where\ S > C\alpha$
$$R_a = R_m + \frac{(S-C(1-\alpha))}{\gamma} \cdot \left(\frac{(1-\alpha)-\frac{S}{C}}{1-\alpha}\right)^b$$ Eq. (A.11)
$where\ S < C(1-\alpha)$
Where $R_a$ is the actual dam release [m³ s⁻¹], $S$ the dam storage capacity [m³], $\alpha$ the fraction of the capacity
that is the desired maximum [-], $\beta$ the exponent determining the correction increase [-], and $\gamma$ the
parameter determining the period when the release is corrected [s⁻¹]. In testing the exponent and period
were tuned to 0.6 and 5 days respectively.
**5.4     Appendix D: Water demand**
**5.4.1     Fitting and validation data**
Data on irrigation, domestic, and industrial water withdrawals were based on the AQUASTAT database
(FAO, 2016), EUROSTAT database (EC, 2019) and United Nations World Water Development Report
(Connor, 2015). Data on GDP per capita and GVA were abstracted from the Maddison Project Database
2018 (Bolt et al., 2018), Penn World Table 9.0 (Feenstra et al., 2015) and World Bank Development
Indicators (World bank, 2010).
Available data for domestic an industrial withdrawals were divided into a dataset used for parameter
fitting (80 %) and a dataset used for validation (20 %). Domestic water demands were estimated for
each United Nations sub-region, and thus the data was divided per sub-region to ensure a good global
coverage of data. In the same manner industrial water demand were divided per country. In case there
is only a single data entry, the entry was added to both the fitting and validation data.
**5.4.2     Irrigation sector**
Conventional irrigation demands were calculated when soil moisture contents drop below the critical
threshold where evapotranspiration will be limited. Demands were set to relieve water stress (Eq. A.12).
Paddy irrigation demands were set to always keep the soil moisture content of the upper soil layer
saturated (Eq. A.13), similar to Hanasaki et al. (2008b) and Wada et al. (2014). For paddy irrigation, the
saturated hydraulic conductivity of the upper soil layer was reduced by its cubed root to simulate
puddling practices, as recommended by the CROPWAT model (Smith, 1996). Total irrigation demands
were adjusted by the irrigation efficiency (Eq. A.14). Paddy irrigation used an irrigation efficiency of 1
since the water losses were already incorporated in the water demand calculation.
$ID'_{conventional} = (W_{cr,1} + W_{cr,2}) - (W_1 + W_2)$          Eq. (A.12)

$where\ W_1 + W_2 < W_{cr,1} + W_{cr,2}$

$ID'_{paddy} = W_{max,1} - W_1$          Eq. (A.13)

$where\ W_1 < W_{max,1}$

$ID = ID' * IE$          Eq. (A.14)
Where $ID'_{conventional}$ is the conventional crop irrigation demand [mm], $ID'_{paddy}$ is the paddy crop irrigation
demand [mm], $ID$ is the total irrigation demand [mm], $W_1$ and $W_2$ are the soil moisture contents of the
first and second soil layer respectively [mm], $W_{cr}$ is the critical soil moisture content [mm], $W_{max}$ the
maximum soil moisture content [mm], and $IE$ is the irrigation efficiency [mm mm$^{-1}$].
**5.4.3   Domestic sector**
Domestic water demands were represented by using a sigmoid curve for the calculation of structural
domestic water demands (Eq.A.15) and a efficiency rate for the calculation of water-use efficiency
increases (Eq. A.16). These equations differ slightly from Alcamo et al. (2003) since our study used the
base 10 logarithms of GDP and water withdrawals per capita as they provided a better fit.
$DSW_y = DSW_{min} + (DSW_{max} - DSW_{min}) * \dfrac{1}{1 + e^{-f(GDP_y - o)}}$      Eq. (A.15)
$DW_y = 10^{DSW_y} \cdot TE^{y - y_{base}}$          Eq. (A.16)
Where $DSW$ is the yearly structural domestic withdrawal [log10 m$^3$ cap$^{-1}$], $DW$ the yearly domestic
withdrawal [m$^3$ cap$^{-1}$], $DSW_{min}$ the minimum structural domestic withdrawal [log10 m$^3$ cap$^{-1}$], $DSW_{max}$
the maximum structural domestic withdrawal (without technological improvement) [log10 m$^3$ cap$^{-1}$],
*GDP* the yearly gross domestic product [log10 USD$_{equivalent}$ cap$^{-1}$], *f* [-] and *o* [log10 USD$_{equivalent}$] the
parameters that determine the range and steepness of the sigmoid curve, *y* the year index, *TE* the
technological efficiency rate [-], and $y_{base}$ the base year (taken to be 1980).
*DW$_{min}$* was set at 7.5 l cap$^{-1}$ d$^{-1}$ based on the World Health Organisation standard (Reed and Reed, 2013),
*DW$_{max}$* was estimated at around 450 l cap$^{-1}$ y$^{-1}$ based on a global curve fit, and *TE* was set at 0.995, 0.99,
and 0.98 for developing, transition and developed countries respectively (United Nations development
status classification) based on Flörke et al. (2013). Curve parameters f and o were estimated for the 23
United Nations sub-regions based on the GDP per capita and domestic water withdrawal data. In case
insufficient data was available to calculate parameters values, regional (4 sub-regions) or global (4 sub-
regions) parameter estimates were used.

### 5.4.4    Industrial sector

Industrial water demands were represented by using a linear formula for the calculation of structural
industrial water demands (Eq. A.17) and a efficiency rate for the calculation of water-use efficiency
increases (Eq. A.18).
$$ISW_y = ISW_{int} \cdot GVA_y \qquad \text{Eq. (A.17)}$$
$$IW_y = \text{ISW}_y \cdot TE^{y - y_{base}} \qquad \text{Eq. (A.18)}$$
Where *ISW* is the yearly structural industrial withdrawal [m$^3$], *ISW$_{int}$* the country specific industrial water
intensity [m USD$_{equivalent}$$^{-1}$], *IW* the yearly industrial withdrawal [m$^3$], *GVA* the yearly gross value added
by industry [USD$_{equivalent}$], *y* the year index, $y_{base}$ the base year (taken to be the year when the industrial
water intensity is determined), and *TE* the technological efficiency rate [-].
*TE* was set at 0.976 and 1 for OECD and non-OECD countries respectively before the year 1980, 0.976
between the years 1980 and 2000 and 0.99 after the year 2000 based on Flörke et al. (2013). Industrial
water intensities were estimated for the 246 United Nations countries based on the GVA and industrial
water withdrawal data. In case insufficient data was available to calculate the industrial water intensities,
either sub-regional (56 countries), regional (17 countries) or global (9 countries) intensities estimates
were used.

### 5.4.5 Energy sector

For each thermoelectric power plant the water intensity was combined with their generation to calculate the water demands (Eq. A.19). Actual generation is estimated by adjusting the installed generation capacity by 46 % for fossil, 72 % for nuclear, and 56 % for biomass power plants (based on EIA national annual generation data (EIA, 2013))

$$EW_y = EW_{int} \cdot G_y \qquad\qquad \text{Eq. (A.19)}$$

Where $EW$ is the yearly energy withdrawal [$m^3$], $EW_{int}$ the energy water intensity [$m^3$ $MWh^{-1}$], $G$ the yearly generation for each plant [MWh], and $y$ the year index.

The energy water demands were subtracted from the industrial water demands at the location of each power plant. In cases where the grid cell industrial water demand was less than the energy water demand, national industrial water demands were lowered. In cases where even the national industrial water demands were less than the national energy water demand (3 countries), the energy water demands were lowered instead. Energy demands were lowered until 10 % of the national industrial water demand remains, to ensure some spatial coverage of industrial and energy water demands.

### 5.4.6 Livestock sector

Livestock water demands were estimated by combining the livestock population with the water requirements for each livestock variety (Eq. A.20).

$$LW_y = LW_{int} \cdot L \qquad\qquad \text{Eq. (A.20)}$$

Where $LW$ is the yearly livestock withdrawal [$m^3$], $LW_{int}$ the livestock water intensity [$m^3$ $livestock^{-1}$], $L$ the livestock number for each variety [livestock].

### 5.5 Appendix E: General performance

VIC-WUR monthly discharge and monthly terrestrial total water storage anomalies were compared with observations from the GRDC dataset (GRDC, 2003) and GRACE satellite dataset (NASA, 2002) for eight major river basins (not included in the main text; Fig. A1). Discharge stations were selected if the upstream area was larger than 10000 m2, matched the simulated upstream area at the station location,):

Amazon, Congo, Lena, Volga, Yangtze, Danube, Columbia, and Mississippi river basins. A 300km
gaussian filer has been applied to the total water storage simulation data (similar to Long et al. (2015)).

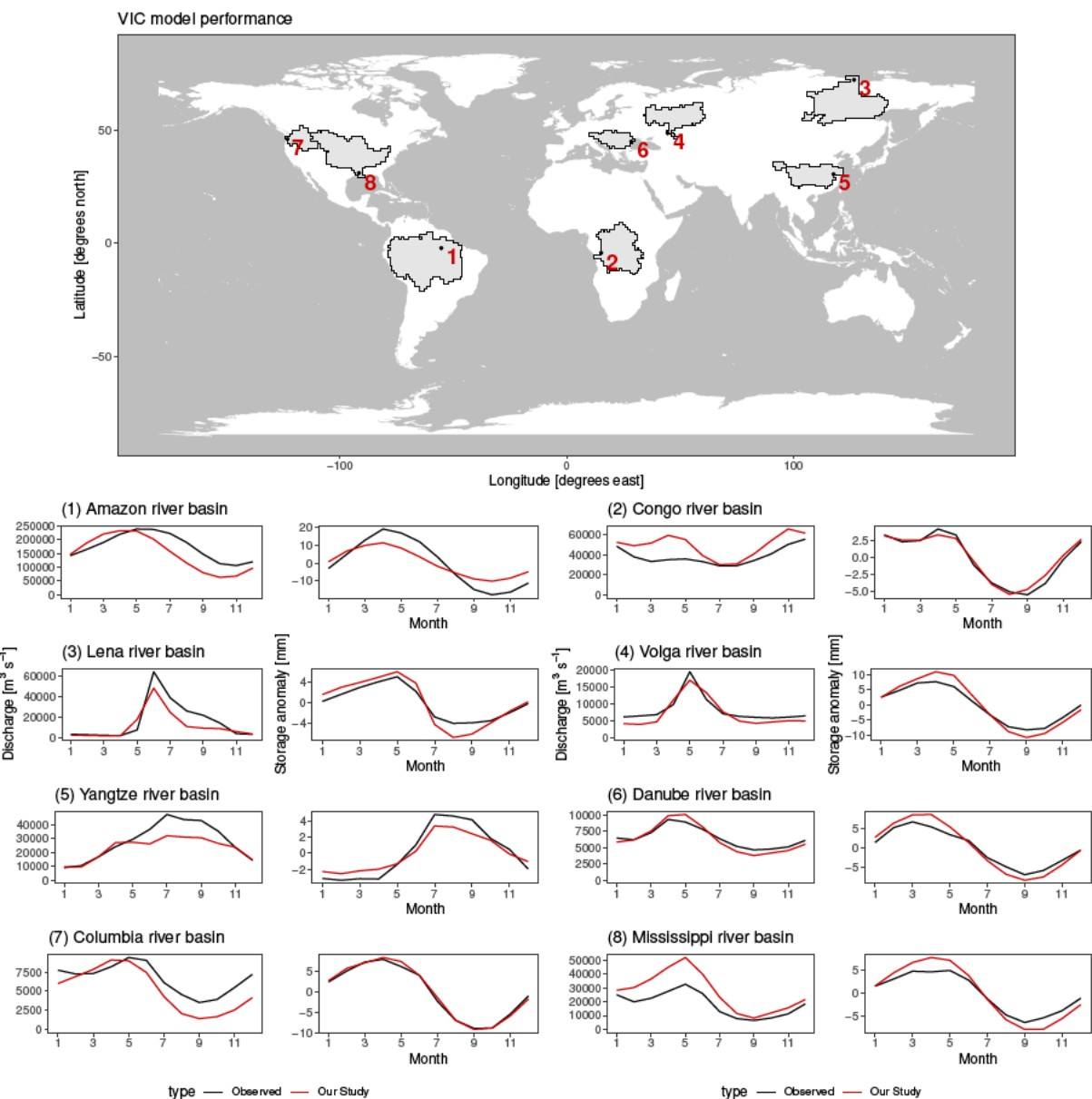


**Figure A1: Comparison between simulated and GRDC and GRACE observed discharge and terrestrial total water storage anomalies. Figures indicate multi-year averages of human-impacted simulations (red) and observations (black).**

## 6    Code availability


All code for the VIC-WUR model is freely available at github.com/wur-wsg/VIC (tag VIC-WUR.2.1.0;
DOI 10.5281/zenodo.3934325) under the GNU General Public License, version 2 (GPL-2.0). VIC-
WUR documentation can be found at vicwur.readthedocs.io. The original VIC model is freely available
at github.com/UW-Hydro/VIC (tag VIC.5.0.1; DOI 10.5281/zenodo.267178) under the GNU General
Public License, version 2 (GPL-2.0). VIC documentation can be found at vic.readthedocs.io.
Documentation and scripts concerning input data used in our study is freely available at
github.com/bramdr/VIC_support (tag VIC-WUR.2.1.0; DOI 10.5281/zenodo.3934363) under the GNU
General Public License, version 3 (GPL-3.0).

## 7 Author contribution

Bram Droppers and Wietse H.P. Franssen developed and tested the model additions introduced in VIC-
WUR. Bram Droppers generated and analysed the results. Michelle T.H. van Vliet, Bart Nijssen and
Fulco Ludwig provided overall oversight and guidance. Bram Droppers prepared the manuscript with
contributions from all co-authors.

## 8 Competing interests

The authors declare that they have no conflict of interest.

## 9 Acknowledgements

We would like to thank Rik Leemans for his guidance and detailed comments. We would like to thank
the Wageningen Institute for Environment and Climate Research (WIMEK) for providing funding for
our research.

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
