# Peer review of "Simulating human impacts on global water resources using"

_Geoscientific Model Development, 2019_

## Referee Comment (RC1) · Anonymous Referee #1 · 22 Jan 2020

General comments

The authors incorporated several water management components, namely, reservoir operation and water requirement estimation for irrigation, livestock, manufacturing, thermal powerplant cooling, domestic use, and environmental flow into the VIC-5 global hydrological model. They compared their simulation results with those of other similar models.

I think this manuscript is excellently written as a model description paper. As for the model itself, however, I feel it includes too few novel aspects. The global offline simulation of VIC was first conducted about 20 years ago (Nijssen et al. 2001). Reservoir operation and irrigation were first introduced in VIC about 15 years ago (Haddeland et al. 2006). The water management components incorporated in this study are mostly

taken from earlier studies (e.g. Alcamo et al. 2003; Hanasaki et al. 2006; Pastor et al. 2014). I understand that this journal does not necessarily require concrete scientific advances, but I personally think that this paper would become better if the authors further emphasize the originality and strength. It is also strongly recommended to provide more concrete information on the capability of this model. In particular, the simulation results should be more rigorously compared with observation, not simulation results of other models.

Specific comments

Line 54 "Several models do not yet incorporate all aspects of anthropogenic water withdrawals...": Some models include 'most' of them already (Döll et al., 2014; Wada et al., 2014; Hanasaki et al., 2018). What is the point here?

Line 89-95: Over all, I feel that the motivation of this study is not well expressed. The present form only tells that the authors want to develop a water resources model based on VIC-5. Perhaps the authors were motivated to integrate the past major works on water management and upgrade the entire model. If this is the case, the model description paper of PCR-GLOBWB2 (Sutanudjaja et al. 2018) provides a good example how to write this part.

Line 227 "Irrigation demands": Does this model support multiple cropping? This point is worth mentioning since it substantially influences irrigation water estimates in Asia, and eventually the globe.

Line 238 "who estimated the irrigation efficiency for 22 United Nations sub-regions based on differences between calculated irrigation requirements and reported irrigation withdrawals": Taking at face value, any calculated requirements will perfectly match with reported withdrawals by this method, which sounds a bit odd. Anyway, irrigation efficiency is quite sensitive to the results and performance, please elaborate the background and concept.

Line 314 "The VIC-WUR model results were compared to several of the ISIMIP simulation round 2a global hydrological impact models": I think VIC-WUR model should be first compared with observation more rigorously. For instance, the simulated river discharge, terrestrial storage components, and reservoir components should be compared with river gauge, terrestrial water storage of the GRACE satellite estimation, and in-situ reservoir operation records, respectively (e.g. Döll et al. 2014; Hanasaki et al. 2018). All the model simulations contain error, hence model-model comparison is not helpful to understand the strength and capability of VIC-WUR.

Line 334 "while the ensemble mean potential and actual withdrawals were only 2200km3 and 1400km3 respectively": According to Figure 3, the potential withdrawal looks more than 2200 km3. Please revisit the number (or figure).

Figure 5: First, domestic water withdrawal of the H08 model is an apparent outlier. It would only make sense if the model reports water consumption, not water withdrawal. Anyway, this figure only tells us that all the models and estimates are different. It doesn't provide any concrete information how well the performance of VIC-WUR is.

Line 400 "Actual irrigation withdrawals of VIC-WUR are high compared to the other models...": The 'actual irrigation withdrawals' simulated by global hydrological models are highly dependent on the model components (e.g. groundwater, small irrigation reservoir, aqueducts, etc.) and the settings (e.g. calculation interval, assignment of environmental flow, etc.). Superficial comparison of numbers is simply meaningless. If the authors wish to keep this part, intensively discuss what can (and cannot) be learned from this intercomparison.

Line 420-434 "When adhering to EFRs the global water withdrawals are reduced substantially...": It is hard for me to support the claim here. The Environmental Flow Requirement (EFR) is, unfortunately, seldom taken care in water scarce regions. If it was taken care, we would observe no groundwater depletion, no terminal lake shrinkage, no flow depletion at river mouth at any places in the world. In reality, we do

observe such 'tragedy' at many places in the world (e.g. the groundwater depletion in the Central Valley in USA, the shrinkage of the Aral Sea, almost complete depletion at the river mouth of the Colorado River). I feel that EFR brings only uncertainties in the phase of model validation, hence better to put aside in a model description paper.

Line 436-448 "However, there are some challenges when applying the methods as described in our paper to future water-food-energy nexus assessments": I am not totally sure whether this paragraph is necessary in this paper. Indeed, the nexus has been extensively studied in the last decade, and some studies have already addressed some of the questions the authors raised. For instance, the community of integrated assessment models have studied on water scarcity on energy generation and manufacturing (Hejazi et al. 2014; Fujimori et al., 2017; Bijl et al. 2018).

References

Alcamo, J., Döll, P., Henrichs, T., Kaspar, F., Lehner, B., Rösch, T., and Siebert, S.: Development and testing of the WaterGAP 2 global model of water use and availability, Hydrological Sciences Journal-Journal Des Sciences Hydrologiques, 48, 317-337, 10.1623/hysj.48.3.317.45290, 2003.

Bijl, D. L., Biemans, H., Bogaart, P. W., Dekker, S. C., Doelman, J. C., Stehfest, E., and van Vuuren, D. P.: A Global Analysis of Future Water Deficit Based on Different Allocation Mechanisms. Water Resources Research, 54(8), 5803-5824. https://doi.org/10.1029/2017WR021688, 2018

Döll, P., Müller Schmied, H., Schuh, C., Portmann, F. T., and Eicker, A.: Global-scale assessment of groundwater depletion and related groundwater abstractions: Combining hydrological modeling with information from well observations and GRACE satellites, Water Resources Research, 50, 5698-5720, 10.1002/2014wr015595, 2014.

Fujimori, S., Hanasaki, N., and Masui, T.: Projections of industrial water withdrawal under shared socioeconomic pathways and climate mitigation scenarios, Sustainability

Science, 12, 275-292, 10.1007/s11625-016-0392-2, 2017.

Haddeland, I., Skaugen, T., and Lettenmaier, D. P.: Anthropogenic impacts on continental surface water fluxes, Geophys. Res. Lett., 33, L08406, doi:10.1029/2006GL026047, 2006.

Hanasaki, N., Kanae, S., and Oki, T.: A reservoir operation scheme for global river routing models, Journal of Hydrology, 327, 22-41, 10.1016/j.jhydrol.2005.11.011, 2006.

Hanasaki, N., Yoshikawa, S., Pokhrel, Y., and Kanae, S.: A global hydrological simulation to specify the sources of water used by humans, Hydrol. Earth Syst. Sci., 22, 789-817, 10.5194/hess-22-789-2018, 2018.

Hejazi, M., Edmonds, J., Clarke, L., Kyle, P., Davies, E., Chaturvedi, V., Wise, M., Patel, P., Eom, J., Calvin, K., Moss, R., and Kim, S.: Long-term global water projections using six socioeconomic scenarios in an integrated assessment modeling framework, Technological Forecasting and Social Change, 81, 205-226, http://dx.doi.org/10.1016/j.techfore.2013.05.006, 2014.

Nijssen, B., O'Donnell, G. M., Lettenmaier, D. P., Lohmann, D., and Wood, E. F.: Predicting the discharge of global rivers, Journal of Climate, 14, 3307-3323, Available at http://www.ce.washington.edu/pub/HYDRO/nijssen/vic_global/index.html, 2001.

Pastor, A. V., Ludwig, F., Biemans, H., Hoff, H., and Kabat, P.: Accounting for environmental flow requirements in global water assessments, Hydrol. Earth Syst. Sci., 18, 5041-5059, 10.5194/hess-18-5041-2014, 2014.

Sutanudjaja, E. H., van Beek, R., Wanders, N., Wada, Y., Bosmans, J. H. C., Drost, N., van der Ent, R. J., de Graaf, I. E. M., Hoch, J. M., de Jong, K., Karssenberg, D., López López, P., Peßenteiner, S., Schmitz, O., Straatsma, M. W., Vannametee, E., Wisser, D., and Bierkens, M. F. P.: PCR-GLOBWB 2: a 5 arcmin global hydrological and water resources model, Geosci. Model Dev., 11, 2429-2453, 10.5194/gmd-11-2429-2018, 2018.

Wada, Y., Wisser, D., and Bierkens, M. F. P.: Global modeling of withdrawal, allocation and consumptive use of surface water and groundwater resources, Earth Syst. Dynam., 5, 15-40, 10.5194/esd-5-15-2014, 2014.

---

## Referee Comment (RC2) · Anonymous Referee #2 · 3 Feb 2020

The manuscript describes in thorough details the integration of models that enhance VIC5 in the representation of anthropogenic activities. The represented human system models include sectoral water demand models, a generic reservoir operations model, and a water supply model. The authors present the overall performance of the model by evaluating the continental water withdrawals with other large-scale hydrology-river routing-water management models. The manuscript is very well structured and written, providing a great resource to support future papers that will likely focus on the analytics instead.

Below are recommendations to further support the individual modules that are being added – some require more details and citations. Another recommendation is about the evaluation of the model performance – only the water withdrawals are evaluated rather

than flow regulation and overall redistribution of water resources and performance of the model in meeting the sectoral demands. The clarifications , and evaluation, are needed in order for this paper to be cited in subsequent papers and support the future analytics.

About the models: Most of the models already exist, whether the hydrology model, the generic operating rules reservoir model, the water supply model, the sectoral water demand models. There is no significant novelty in those individual models. Some models would need to refer to existing models for evaluation:

- The sectoral demand models: (Huang et al., 2018) provides an evaluation of the different water demand models for different sectors. The set up and computed sectoral water demands would need to be further evaluated to support the sectoral water demand models in VIC5-WUR. Please note that the assumption for power plants to run at maximum capacity constantly is not realistic. Capacity factors (ratio of generation over maximum capacity) and generation portfolio are available through the EIA and IEA datasets, which could help improve this demand model, along with suggestions in the other models.

- Enhancement of reservoir releases based on storage levels. (Yassin et al., 2019) and (Rougé et al., 2019) provided a new reservoir operation formulation that modulates releases based on storage levels. While the manuscript is not a review of existing models, the proposed citations should help further support the "model enhancement and improvement" with respect to existing models. Appropriate figures and validation should be provided.

- (Nazemi & Wheater, 2015a, 2015b) provides an overview of existing challenges in large scale water management models. Those papers should be cited in the introduction to complement the authors identified challenges ( the environmental flow) with other identified challenges.

- The supply models: The allocation of sectoral water demand to surface and ground-
water systems as well as the sectoral return flow into the surface water system seems to be equivalent to (Voisin et al., 2017), which should then be cited. The description of how the supply is allocated to the different sectoral water demands needs to be specified in this manuscript. A missing description is how the priority is set between sectoral demands. For example , are thermo-electric plants getting their demand met first before domestic or irrigation demand? Is there any priority for supply allocation based on spatial location? Which grid cells can request water from a mainstream if the main channel is not within this grid cell? Was the Hanasaki et al. (2006) "dependence" database used? While authors indicate that it will be the subject of further research, what is the default implementation that was used in the presented simulations?

About the evaluation to support future studies;

- the introduction is missing a range of large scale studies where such a large scale water management model has been used with the VIC model, albeit not VIC5. While the proposed set up seems more complete, it seems that the paper should still cite those studies as they represent to a certain extent an earlier version of this integrated model (Voisin et al., 2017; Voisin et al., 2018; Zhou, Voisin, Leng, Huang, & Kraucunas, 2018)

- The overall evaluation of the model is limited to continental sectoral water withdrawals. The model is expected to be used in large scale water-energy-food nexus numerical experiments, yet there is no evaluation of the terrestrial water storage with respect to GRACE as performed in other equivalent models valuations, or in flow (Yassin et al., 2019), or in supply deficit metrics as "accounting for supply from unsustainable sources" (Döll et al., 2012) or as unmet demand (Voisin et al., 2013), which allows to evaluate the overall performance of the sectoral water management model.

References

Döll, P., Hoffmann-Dobrev, H., Portmann, F. T., Siebert, S., Eicker, A., Rodell, M., . . . Scanlon, B. R. (2012). Impact of water withdrawals from groundwater and surface

water on continental water storage variations. Journal of Geodynamics, 59-60, 143-156. doi:https://doi.org/10.1016/j.jog.2011.05.001

Huang, Z., Hejazi, M., Li, X., Tang, Q., Vernon, C., Leng, G., . . . Wada, Y. (2018). Reconstruction of global gridded monthly sectoral water withdrawals for 1971–2010 and analysis of their spatiotemporal patterns. Hydrol. Earth Syst. Sci., 22(4), 2117-2133. doi:10.5194/hess-22-2117-2018

Nazemi, A., & Wheater, H. S. (2015a). On inclusion of water resource management in Earth system models – Part 1: Problem definition and representation of water demand. Hydrol. Earth Syst. Sci., 19(1), 33-61. doi:10.5194/hess-19-33-2015

Nazemi, A., & Wheater, H. S. (2015b). On inclusion of water resource management in Earth system models – Part 2: Representation of water supply and allocation and opportunities for improved modeling. Hydrol. Earth Syst. Sci., 19(1), 63-90. doi:10.5194/hess-19-63-2015

Rougé, C., Reed, P. M., Grogan, D. S., Zuidema, S., Prusevich, A., Glidden, S., . . . . Lammers, R. B. (2019). Coordination and Control: Limits in Standard Representations of Multi-Reservoir Operations in Hydrological Modeling. Hydrol. Earth Syst. Sci. Discuss., 2019, 1-37. doi:10.5194/hess-2019-589

Voisin, N., Hejazi, M. I., Leung, L. R., Liu, L., Huang, M. Y., Li, H. Y., & Tesfa, T. (2017). Effects of spatially distributed sectoral water management on the redistribution of water resources in an integrated water model. Water Resources Research, 53(5), 4253-4270. doi:10.1002/2016wr019767

Voisin, N., Kintner-Meyer, M., Wu, D., Skaggs, R., Fu, T., Zhou, T., . . . Kraucunas, I. (2018). Opportunities for Joint Water–Energy Management: Sensitivity of the 2010 Western U.S. Electricity Grid Operations to Climate Oscillations. Bulletin of the American Meteorological Society, 99(2), 299-312. doi:10.1175/bams-d-16-0253.1

Voisin, N., Liu, L., Hejazi, M., Tesfa, T., Li, H., Huang, M., . . . Leung, L. R. (2013).

[Figure]

One-way coupling of an integrated assessment model and a water resources model: evaluation and implications of future changes over the US Midwest. Hydrology and Earth System Sciences, 17(11), 4555-4575. doi:10.5194/hess-17-4555-2013

Yassin, F., Razavi, S., Elshamy, M., Davison, B., Sapriza-Azuri, G., & Wheater, H. (2019). Representation and improved parameterization of reservoir operation in hydrological and land-surface models. Hydrol. Earth Syst. Sci., 23(9), 3735-3764. doi:10.5194/hess-23-3735-2019

Zhou, T., Voisin, N., Leng, G., Huang, M., & Kraucunas, I. (2018). Sensitivity of Regulated Flow Regimes to Climate Change in the Western United States. Journal of Hydrometeorology, 19(3), 499-515. doi:10.1175/jhm-d-17-0095.1

―――――――――――――――

---

## Referee Comment (RC3) · Anonymous Referee #3 · 6 Mar 2020

General Comments:

The authors enhance the VIC model with several additional modules in an attempt to better capture anthropogenic and environmental flow requirement impacts on water use. Additional modules include those for integrated routing, sectoral water use, EFR for surface and subsurface water systems and dam operations. Overall I believe that this paper was well written and that the modules considered add appropriate value to the VIC model.

However, I believe that the methodology itself lacks in novel advancements and claims of a "first step towards integrated water-food-energy nexus modeling" (line 33) may mislead future readers. Comparisons are made only against other hydrologic models rather than considering historical datasets of observed sectoral and/or global water

withdrawals. In addition, several Integrated Assessment Models have now begun water integration to better understand cross-sectoral impacts on water. While these often are not solved at the resolution of a model such as VIC, they may provide a useful additional comparison.

Specific Comments:

Line 33: "The improvements made here are a first step towards integrated water-food-energy nexus modeling" This conclusion should be clarified to distinguish the fact that this study may provide a first toward towards FEW modeling in hydrologic models. Integrated Assessment Models have increasingly been investigating FEW nexus inter-actions and should be acknowledged within the manuscript (Hejazi et al., 2014, Bijl et al., 2018, Graham et al., 2018, among several others)

Several modules are based on prior work that is now 10-15 years old (Shiklomanov 2000; Goldstein and Smith, 2002; Haddeland et al., 2006). It should be more carefully noted throughout the text the novelty of what is being added to the modeling community.

Line 328: the study is mentioned to use varying socioeconomic predictors. These could be better explained in section 2.3.2 in order to specify where GDP and GVA are obtained.

Lines 406-408: "To our knowledge no previous study has estimated the amount of global non-renewable groundwater withdrawals without using on the the models mentioned above" - see Turner et al. (2019) or Kim et al. (2016) for additional groundwater withdrawal modeling capabilities.

Line 426: "Note that VIC-WUR does not include non-renewable groundwater withdrawals, while these withdrawals would affect baseflow to a lesser degree" - I am confused, then why was there a discussion on about this in paragraph starting at line 400? Maybe consider reorganizing these thoughts.

References:

Bijl, David L., Patrick W. Bogaart, Stefan C. Dekker, and Detlef P. van Vuuren. "Unpacking the nexus: Different spatial scales for water, food and energy." Global Environmental Change 48 (2018): 22-31.

Goldstein, Robert, and W. E. P. R. I. Smith. Water & sustainability (volume 4): US electricity consumption for water supply & treatment-the next half century. Electric Power Research Institute, 2002.

Graham, Neal T., Evan GR Davies, Mohamad I. Hejazi, Katherine Calvin, Son H. Kim, Lauren Helinski, Fernando R. Miralles‐Wilhelm et al. "Water sector assumptions for the Shared Socioeconomic Pathways in an integrated modeling framework." Water Resources Research 54, no. 9 (2018): 6423-6440.

Haddeland, Ingjerd, Thomas Skaugen, and Dennis P. Lettenmaier. "Anthropogenic impacts on continental surface water fluxes." Geophysical Research Letters 33, no. 8 (2006).

Hejazi, Mohamad, James Edmonds, Leon Clarke, Page Kyle, Evan Davies, Vaibhav Chaturvedi, Marshall Wise et al. "Long-term global water projections using six socioeconomic scenarios in an integrated assessment modeling framework." Technological Forecasting and Social Change 81 (2014): 205-226.

Kim, Son H., Mohamad Hejazi, Lu Liu, Katherine Calvin, Leon Clarke, Jae Edmonds, Page Kyle, Pralit Patel, Marshall Wise, and Evan Davies. "Balancing global water availability and use at basin scale in an integrated assessment model." Climatic Change 136, no. 2 (2016): 217-231.

Shiklomanov, Igor A. "Appraisal and assessment of world water resources." Water international 25, no. 1 (2000): 11-32.

Turner, Sean WD, Mohamad Hejazi, Catherine Yonkofski, Son H. Kim, and Page Kyle. "Influence of Groundwater Extraction Costs and Resource Depletion Limits on Simulated Global Nonrenewable Water Withdrawals Over the Twenty‐First Century."

Earth's Future 7, no. 2 (2019): 123-135.

---

## Author Comment (AC1) · 6 Mar 2020

Dear referee,

Thank you very much for reviewing our paper titled "Simulating human water impacts on global water
resources using VIC-5" and for your valuable comments and suggestions. Below we address your
comments (shown in italic), with our responses in blue.

**Model performance**

The referee suggests that we should *"provide more concrete information about the capability of this*
*model. In particular, the simulation results should be more rigorously compared with observation, not*
*simulation results of other models"*. More specifically (as stated in the specific comments), *"river*
*discharge, terrestrial storage components, and reservoir components should be compared with river*
*gauge, terrestrial water storage of the GRACE satellite estimation and in-situ reservoir operation*
*records respectively"*. These suggestions were also raised by the other reviewer.

We agree with these suggestions and we will include a rigorous evaluation of the hydrological model
performance. We will compare model simulations with observations and/or reported data on discharge,
total water storage, reservoir storage and sectoral water demands. The following approaches are
proposed:

1.  Simulated discharge will be compared with monthly timeseries and multi-year average
discharge from the GRDC dataset, between 1980 and 2010. Stations are selected within the
major river basins of the original VIC calibration paper of Nijssen et al. (2001). Naturalized
discharge as well as human-modified discharge simulations will be compared in this manner.
2.  Simulated total water storage will be compared with monthly timeseries, multi-year-average
total water storage and inter-annual water storage trends from the GRACE satellite dataset, for
the period 2004-2016. To do so, a 300km gaussian filter will be applied to the simulated total
water storage, as it is in the GRACE dataset. Total water storage will be compared for the same
river basins as in the discharge comparison. Naturalized and human-modified total water storage
simulations will be compared in this manner. These results will also include the  unmet water
demands, subsequent non-renewable groundwater abstractions and long-term total water
storage exploitation.
3.  Simulated sectoral water demand will be compared with monthly timeseries from the Huang et
al. (2018) dataset. This is in addition to the comparison to the Shiklomanov (2000) dataset and
FAOSTAT (FAO, 2016), EUROSTAT (EC, 2019) and WWDR (Connor, 2015) datasets already
used in the paper. Sectoral water demands will be compared for the world and for the 5 regions

| 34 | | used in this paper (Africa, Americas, Asia, Europe and Oceania); and separately for each sector |
|----|---|---|
| 35 | | (irrigation, domestic, industrial and livestock) separately. |
| 36 | 4. | Simulated reservoir inflow, storage and release will be compared with monthly timeseries from |
| 37 | | Yassin et al. (2019) (assuming this data is shared), Rougé et al. (2019) and Hanasaki et al. (2006) |
| 38 | | datasets. Dams are selected based on data availability and evaluation will focus on large dams. |

**Novelty**

The referee comments that the model *"includes too few novel aspects"*, since the reservoirs and irrigation modules were already included in previous VIC versions and the water management components were taken from several previous studies. The referee also comments that *"this paper would become better if the authors further emphasize the originality and strength"* of the study. Also, the referee feels that *"the motivation of this study is not well expressed"*.

In response to the issue raised by the referee, we will describe the originality and strength of the model, as well as a clear motivation for our study more clearly. We will clearly to acknowledge that the water management modules are based on previous major works, while describing clearly improvements compared to previous VIC studies, as well as other global hydrological modelling studies.

Compared to previous VIC studies, our model study includes the full range of water-use sectors (including domestic, industrial, energy and livestock), which have been estimated independently. Also, the routing module was fully integrated in VIC-5, which was not possible in previous VIC versions. This heavily decreases computation times for human-impact studies and provides a much improved framework for other future human-impact studies. Water-use sectors can also use groundwater as a resources, which directly impacts baseflow and thus downstream (dry-season) water availability. Compared to other studies, environmental flow requirements from surface- and groundwater systems for terrestrial freshwater ecosystems have been fully integrated. In addition, environmental flow requirements for groundwater into a hydrological model is also a novel component.

Concluding, we do not agree that the study includes too few novel aspects. However, we agree a clearer distinction needs to be made between aspects of model development and scientific development in this study. Therefore we will adjust our manuscript in several places.

Lines 84-88: "Several studies used VIC to simulate the anthropogenic impacts of irrigation and dam operation on water resources (Haddeland et al., 2006a; Haddeland et al., 2006b; Zhou et al., 2015; Zhou et al., 2016) based on the model setup of Haddeland et al. (2006b). However, water withdrawals for other sectors and flow requirements for freshwater ecosystems were ignored in these studies"

Will change to: "Several studies used VIC to simulate the worldwide anthropogenic impacts of irrigation and dam operation on water resources (Haddeland et al., 2006a; Haddeland et al., 2006b; Zhou et al., 2015; Zhou et al., 2016) based on the model setup of Haddeland et al. (2006b). However, groundwater withdrawals, water withdrawals for other sectors and flow requirements for freshwater ecosystems were
not included in these studies."

Lines 89-90: "Our study aims to increase the applicability of the VIC-5 model for water resource
assessments, specifically by including human impacts and environmental flow requirements."

Will change to: "Our study aims to increase the applicability of the VIC model for water resource
assessments, specifically by including human impacts and environmental flow requirements."

Line 93: "(...) impacts on water resources. These modules include (...)"

Will change to: "(...) impacts on water resources. These modules will integrate the previous major works
on anthropogenic-impact modelling into VIC-5. modules include (...)"

Line 95: "(...) systems, and dam operation."

Will change to: "(...) systems, and dam operation. While the study of Haddeland et al. (2006b) already
included some offline anthropogenic-impact modules (surface water use for the irrigation sector and
dam operation), the new VIC-5 model structure and integrated routing are better suited for global
integrated water-resource assessments and substantially decreases computation times (see Section 2.1)."

Line 104: "(...) imposed by EFRs."

Will change to: "(...) imposed by EFRs. This EFR assessment is included to indicate the effects of the
newly integrated (groundwater) environmental flow requirements on worldwide water availability. "

**Specific comments**

*"Line 54 "Several models do not yet incorporate all aspects of anthropogenic water withdrawals..."*:

*Some models include 'most' of them already (Döll et al., 2014; Wada et al., 2014; Hanasaki et al.,*

*2018). What is the point here?"*

We agree with the referee that this sentence (and paragraph) may cause some confusion. Therefore we will rewrite this part of the introduction.

Lines 53-56: "However, further advancements are needed to improve the integration of anthropogenic impacts into hydrological models (Döll et al., 2016). Several models do not yet incorporate all aspects of anthropogenic water withdrawals such as domestic, manufacturing and energy (thermoelectric) water withdrawals from both ground and surface water."

Will change to: "Further advancements are needed to improve the integration of anthropogenic impacts into hydrological models (Döll et al., 2016). The VIC model does not yet incorporate all aspects of anthropogenic water withdrawals such as domestic, manufacturing and energy (thermoelectric) water withdrawals from both ground and surface water."

And will move behind line 88.

*"Line 227 "Irrigation demands": Does this model support multiple cropping? This point is worth*

*mentioning since it substantially influences irrigation water estimates in Asia, and eventually the globe"*

Irrigation demands support multiple cropping. This was indirectly described in section 3.1 line 299-300

"MIRCA2000 distinguishes the monthly growing area(s) and season(s) of 26 irrigated and rain-fed crop types around the year 2000" and line 303-304: "Cropland coverage (the cropland area actually growing crops) varied monthly based on the crop growing areas of MIRCA2000. The remainder was treated as bare soil". However, this will be explicitly stated.

Lines 234-235: "(...) applied separately (i.e. in different sub-grids)."

Will change to: "(...) applied separately (i.e. in different sub-grids). Note that multiple cropping seasons are included based on the MIRCA2000 land-use dataset (Portmann et al., 2010)."

*"Line 238 "who estimated the irrigation efficiency for 22 United Nations sub-regions based on*

*differences between calculated irrigation requirements and reported irrigation withdrawals": Taking*

*at face value, any calculated requirements will perfectly match with reported withdrawals by this*

*method, which sounds a bit odd. Anyway, irrigation efficiency is quite sensitive to the results and*

*performance, please elaborate the background and concept."*

The description of the irrigation efficiency implementation will be elaborated upon.

Lines 238-240: "The water loss fraction was based on Frenken and Gillet (2012), who estimated the irrigation efficiency for 22 United Nations sub-regions based on differences between calculated irrigation requirements and reported irrigation withdrawals."

Will change to: "The water loss fraction was based on Frenken and Gillet (2012), who estimated the irrigation efficiency for 22 United Nations sub-regions. Irrigation efficiencies were estimated based on the differences between the calculated crop water requirements (crop evapotranspiration; consumptive water use) and the reported irrigation water withdrawals (including transportation and application losses). Crop water requirements are estimated based on the FAO Irrigation and Drainage paper (Allen et al., 1998). Low irrigation efficiencies can result in irrigation water withdrawals up to four times higher than the crop water requirements in regions such as east- and west Africa."

*"Line 334 "while the ensemble mean potential and actual withdrawals were only 2200km3 and 1400km3 respectively": According to Figure 3, the potential withdrawal looks more than 2200 km3. Please revisit the number (or figure)."*

The number in the text should be 2460 km3.

Lines 333-335: "Annual potential and actual irrigation withdrawals for VIC-WUR were around 3060 km$^3$ and 1870 km$^3$ respectively, while the ensemble mean potential and actual withdrawals were only 2200 km$^3$ and 1400 km$^3$ respectively"

Will change to: "Annual potential and actual irrigation withdrawals for VIC-WUR were around 3060 km$^3$ and 1870 km$^3$ respectively, while the ensemble mean potential and actual withdrawals were only 2460 km$^3$ and 1400 km$^3$ respectively"

*"Figure 5: First, domestic water withdrawal of the H08 model is an apparent outlier. It would only make sense if the model reports water consumption, not water withdrawal. Anyway, this figure only tells us that all the models and estimates are different. It doesn't provide any concrete information how well the performance of VIC-WUR is."*

The data for H08 is the actual domestic water withdrawal as supplied to the ISIMIP2a project. However, to avoid confusion we will remove the model from the analysis of non-irrigation water withdrawals.

The figure was also meant to place the VIC-WUR model in context of the other models. Note that the Shiklomanov (2000) values are based on worldwide reported data (not modelled). However, to provide more concrete information about the performance of VIC-WUR we will compare the model results to Huang et al. (2018), in addition to Shiklomanov (2000) (as described above).

Line 320-321: "H08 additionally provided data for the domestic sector, and PCR-GLOBWB
additionally provided data for the domestic and livestock sector."

Will change to: "PCR-GLOBWB additionally provided data for the domestic and livestock sector."

*"Line 400 "Actual irrigation withdrawals of VIC-WUR are high compared to the other Models...": The*
*'actual irrigation withdrawals' simulated by global hydrological models are highly dependent on the*
*model components (e.g. groundwater, small irrigation reservoir, aqueducts, etc.) and the settings (e.g.*
*calculation interval, assignment of environmental flow, etc.). Superficial comparison of numbers is*
*simply meaningless. If the authors wish to keep this part, intensively discuss what can (and cannot) be*
*learned from this intercomparison."*

The referee indicates that, without a proper description of the model setup, comparison between different
model results is meaningless. Therefore, we will describe most of the model settings and components
as well as more rigorously discuss the model differences in the results. Also, we will compare the model
results to the worldwide gridded sectoral water withdrawal data of Huang et al. (2018). However, we
would still like to include these results since it puts VIC-WUR in the context of the older VIC version
of Haddeland et al. (2006b) and other global hydrological models.

The results indicate to what extent the hydrological models are able to use renewable water resources
for the anthropogenic water demand (and thus to what extend there would be non-renewable water
withdrawals). Also, there is no other way to compare the water resource availability on a global scale,
since such observations are not available.

Line 317-318: "(...) and WaterGAP (Muller Schmied et al., 2016). The ISIMIP2a outputs (...)"

Will change to: "(...) and WaterGAP (Muller Schmied et al., 2016). For simulation round 2a the models
were required to harmonize their land-use and weather-forcing inputs. Also, no non-renewable water
abstractions were allowed, as not to violate the water balance. Of these models only PCR-GLOBLWB
includes (renewable) groundwater withdrawals and only the VIC model did not consider paddy rice
practices. The ISIMIP2a outputs (...)"

*"Line 420-434 "When adhering to EFRs the global water withdrawals are reduced substantially...": It*
*is hard for me to support the claim here. The Environmental Flow Requirement (EFR) is, unfortunately,*
*seldom taken care in water scarce regions. If it was taken care, we would observe no groundwater*
*depletion, no terminal lake shrinkage, no flow depletion at river mouth at any places in the world. In*
*reality, we do observe such 'tragedy' at many places in the world (e.g. the groundwater depletion in the*
*Central Valley in USA, the shrinkage of the Aral Sea, almost complete depletion at the river mouth of*

*the Colorado River). I feel that EFR brings only uncertainties in the phase of model validation, hence*
*better to put aside in a model description paper."*

We did not try to imply that Environmental Flow Requirements (EFRs) are seldom taken care of, rather
that the opposite is true. However, since the integrated surface and groundwater EFRs are some of the
additions to the hydrological model, we think it wise to discuss some of the impacts of this addition and
its implications. However, the discussion will be shortened.

Line 351-352: "Therefore, the impact of the environmental flow requirements was largest in
groundwater dependent regions"

Will change to: "Therefore, the potential impact of the environmental flow requirements (if adhered to)
would be largest in groundwater dependent regions"

Line 420-421: "When adhering to EFRs the global water withdrawals are reduced substantially,
especially due to groundwater withdrawal limitations"

Will change to: "If water-users would adhere to EFRs the global water withdrawals reduce substantially,
especially due to constrains in groundwater withdrawals"

Lines 421-425: "This limitation indicates competition between water allocated for anthropogenic uses
and environmental purposes. In addition, groundwater withdrawal reductions upstream lead to increased
surface water availability downstream. This interaction results in a trade-off between upstream
groundwater withdrawals and downstream surface water withdrawals."

Will be removed

*"Line 436-448 "However, there are some challenges when applying the methods as described in our*
*paper to future water-food-energy nexus assessments": I am not totally sure whether this paragraph is*
*necessary in this paper. Indeed, the nexus has been extensively studied in the last decade, and some*
*studies have already addressed some of the questions the authors raised. For instance, the community*
*of integrated assessment models have studied on water scarcity on energy generation and*
*manufacturing (Hejazi et al. 2014; Fujimori et al., 2017; Bijl et al. 2018)."*

We agree with the reasoning of the referee. This section takes up too much space in the discussion
section and we will therefore remove this paragraph.

We hope the referee agrees with our changes made, and are open to any further suggestions or comments.

Sincerely,

Bram Droppers on behalf of all co-authors

**References**

Allen, R. G., Pereira, L. S., Raes, D., and Smith, M.: Crop Evapotranspiration - Guidelines for computing crop water requirements, Food and Agricultural Organisation, Rome, Italy, 326, 1998.

Connor, R.: Water for a sustainable world, United Nations Educational, Scientific and Cultural Organisation, Paris, France, 139, 2015.

Döll, P., Douville, H., Guntner, A., Muller Schmied, H., and Wada, Y.: Modelling Freshwater Resources at the Global Scale: Challenges and Prospects, Surv Geophys, 37, 195-221, 10.1007/s10712-015-9343-1, 2016.

Frenken, K., and Gillet, V.: Irrigation water requirement and water withdrawal by country, Food and agricultural organisation, Rome, Italy, 264, 2012.

Haddeland, I., Lettenmaier, D. P., and Skaugen, T.: Effects of irrigation on the water and energy balances of the Colorado and Mekong river basins, J Hydrol, 324, 210-223, 10.1016/j.jhydrol.2005.09.028, 2006a.

Haddeland, I., Skaugen, T., and Lettenmaier, D. P.: Anthropogenic impacts on continental surface water fluxes, Geophys Res Lett, 33, 10.1029/2006gl026047, 2006b.

Hanasaki, N., Kanae, S., and Oki, T.: A reservoir operation scheme for global river routing models, J Hydrol, 327, 22-41, 10.1016/j.jhydrol.2005.11.011, 2006.

Huang, Z., Hejazi, M., Li, X., Tang, Q., Vernon, C., Leng, G., Liu, Y., Döll, P., Eisner, S., Gerten, D., Hanasaki, N., and Wada, Y.: Reconstruction of global gridded monthly sectoral water withdrawals for 1971–2010 and analysis of their spatiotemporal patterns, Hydrol. Earth Syst. Sci., 22, 2117-2133, 10.5194/hess-22-2117-2018, 2018.

Muller Schmied, H., Adam, L., Eisner, S., Fink, G., Flörke, M., Kim, H., Oki, T., Portmann, F. T., Reinecke, R., Riedel, C., Song, Q., Zhang, J., and Döll, P.: Variations of global and continental water balance components as impacted by climate forcing uncertainty and human water use, Hydrol Earth Syst Sc, 20, 2877-2898, 10.5194/hess-20-2877-2016, 2016.

Nijssen, B., O'Donnell, G. M., Lettenmaier, D. P., Lohmann, D., and Wood, E. F.: Predicting the discharge of global rivers, J Climate, 14, 3307-3323, Doi 10.1175/1520-0442(2001)014<3307:Ptdogr>2.0.Co;2, 2001.

Portmann, F. T., Siebert, S., and Döll, P.: MIRCA2000-Global monthly irrigated and rainfed crop areas around the year 2000: A new high-resolution data set for agricultural and hydrological modeling, Global Biogeochem Cy, 24, 10.1029/2008gb003435, 2010.

Rougé, C., Reed, P. M., Grogan, D. S., Zuidema, S., Prusevich, A., Glidden, S., Lamontagne, J. R., and Lammers, R. B.: Coordination and Control: Limits in Standard Representations of Multi-Reservoir Operations in Hydrological Modeling, Hydrol. Earth Syst. Sci. Discuss., 2019, 1-37, 10.5194/hess-2019-589, 2019.

Shiklomanov, I. A.: Appraisal and assessment of world water resources, Water Int, 25, 11-32, Doi 10.1080/02508060008686794, 2000.

Yassin, F., Razavi, S., Elshamy, M., Davison, B., Sapriza-Azuri, G., and Wheater, H.: Representation and improved parameterization of reservoir operation in hydrological and land-surface models, Hydrol. Earth Syst. Sci., 23, 3735-3764, 10.5194/hess-23-3735-2019, 2019.

Zhou, T., Haddeland, I., Nijssen, B., and Lettenmaier, D. P.: Human induced changes in the global water cycle, AGU Geophysical Monograph Series, Submitted, 2015.

Zhou, T., Nijssen, B., Gao, H. L., and Lettenmaier, D. P.: The Contribution of Reservoirs to Global Land Surface Water Storage Variations, J Hydrometeorol, 17, 309-325, 10.1175/Jhm-D-15-0002.1, 2016.

---

## Author Comment (AC2) · 6 Mar 2020

Dear referee,

Thank you very much for reviewing our paper titled "Simulating human water impacts on global water resources using VIC-5" and for your valuable comments and suggestions. Below we address your comments (shown in italic), with our responses in blue.

**Model performance**

The referee suggests that we should further evaluate model performance, such as *"flow regulation and*

*overall redistribution of water resources and performance of the model in meeting sectoral demands"*.

Later it is mentioned with respect to water stores and/or fluxes: *"there is no evaluation of the terrestrial*

*water storage with respect to GRACE as performed in other equivalent models valuations, or in flow*

*(Yassin et al., 2019), or in supply deficit metrics as "accounting for supply from unsustainable sources"*

*(Döll et al., 2012) or as unmet demand (Voisin et al., 2013), which allows to evaluate the overall*

*performance of the sectoral water management model"*, and with respect to sectoral water demands:

*"The sectoral demand models: (Huang et al., 2018) provides an evaluation of the different water*

*demand models for different sectors. The set up and computed sectoral water demands would need to*

*be further evaluated to support the sectoral water demand models in VIC5-WUR"*, and with respect to reservoir operation: *"Appropriate figures and validation should be provided"*. These suggestions were also addressed by the other reviewer.

We agree with these suggestions and will include a rigorous evaluation of the hydrological model performance. More specifically we will compare model simulations with observations and/or reported data on discharge, total water storage, reservoir storage and sectoral water demands. The following approaches are proposed:

1. Simulated discharge will be compared with monthly timeseries and multi-year average discharge from the GRDC dataset, between 1980 and 2010. Stations are selected within the major river basins of the original VIC calibration paper of Nijssen et al. (2001b). Naturalized discharge as well as human-modified discharge simulations will be compared in this manner.

2. Simulated total water storage will be compared against monthly timeseries, multi-year-average total water storage and inter-annual water storage trends from the GRACE satellite dataset, between 2004 and 2016. To do so, a 300km gaussian filter will be applied to the simulated total water storage, as it is in the GRACE dataset. Total water storage will be compared for the same river basins as in the discharge comparison. Naturalized discharge as well as human-modified total water storage simulations will be compared in this manner. These results will also include the unmet water demands, subsequent non-renewable groundwater abstractions and long-term total water storage exploitation.

3. Simulated sectoral water demand will be compared with monthly timeseries from the Huang et al. (2018) dataset. This is in addition to the comparison to the Shiklomanov (2000) dataset and FAOSTAT (FAO, 2016), EUROSTAT (EC, 2019) and WWDR (Connor, 2015) datasets already used in the paper. Sectoral water demands will be compared for the world and for the 5 regions used in this paper (Africa, Americas, Asia, Europe and Oceania); and separately for each sector (irrigation, domestic, industrial and livestock) separately.

4. Simulated reservoir inflow, storage and release will be compared with monthly timeseries from Yassin et al. (2019) (assuming this data is shared), Rougé et al. (2019) and Hanasaki et al. (2006) datasets. Dams are selected based on data availability and evaluation will focus on large dams.

**Specific comments**

*"Please note that the assumption for power plants to run at maximum capacity constantly is not realistic. Capacity factors (ratio of generation over maximum capacity) and generation portfolio are available through the EIA and IEA datasets"*

Thanks for this comment. Capacity factors on a per-plant basis as mentioned by the referee are not fully available to us, unfortunately. Country-based analysis, based on the EIA dataset, shows that the capacity factors vary per country (fossil: between 1% and 73%; nuclear: between 37% and 88%; biomass: between 15% and 100%) and over time (fossil: between 44% and 48%; nuclear: between 56% and 82%; biomass: between 53% and 58%). These factors may also be cooling system dependent. Due to these data limitations we will use a global mean factor of 46% for fossil, 72% for nuclear and 56% for biomass based power plants.

Line 669-671: "Since there was no observed data about the actual annual generation, each plant was assumed to be running at its installed generation capacity throughout the year, similar to van Vliet et al. (2016)."

Will change to: "Actual generation is estimated by adjusting the installed generation capacity by 46% for fossil, 72% for nuclear and 56% for biomass power plants (based on country-based data of the EIA (EIA, 2013))."

Line 677-681: "In cases where even the national industrial water demands were less than the national energy water demand (5 countries), the energy water demands were lowered instead. This could be the case in countries where power plants do not operate at their installed capacity, as globally around 45% of the installed capacity is actually generated (based on data of van Vliet et al. (2016))."

Will change to: "In cases where even the national industrial water demands were less than the national energy water demand (4 countries), the energy water demands were lowered instead."

*"Enhancement of reservoir releases based on storage levels. (Yassin et al., 2019) and (Rougé et al.,*
*2019) provided a new reservoir operation formulation that modulates releases based on storage levels.*
*While the manuscript is not a review of existing models, the proposed citations should help further*
*support the "model enhancement and improvement" with respect to existing models"*

We have included the citations mentioned by the referee, which also describe generic dam operation
schemes developed for large-scale hydrological modelling.

Line 197-199: "Due to the lack of globally available information on local dam operations, several
generic dam operation schemes were developed for macro-scale hydrological models to reproduce the
effect of dams on natural streamflow (Haddeland et al., 2006; Hanasaki et al., 2006; Zhao et al., 2016)"

Will change to: "Due to the lack of globally available information on local dam operations, several
generic dam operation schemes were developed for macro-scale hydrological models to reproduce the
effect of dams on natural streamflow (Haddeland et al., 2006; Hanasaki et al., 2006; Zhao et al., 2016;
Rougé et al., 2019; Yassin et al., 2019)"

*"(Nazemi & Wheater, 2015a, 2015b) provides an overview of existing challenges in large scale water*
*management models. Those papers should be cited in the introduction to complement the authors*
*identified challenges ( the environmental flow) with other identified challenges"*

We have included the citations mentioned by the referee, as well as Pokhrel et al. (2016) to include a
wider range of review papers that identify the challenges in large-scale hydrological modelling.

Lines 53-54: "However, further advancements are needed to improve the integration of anthropogenic
impacts into hydrological models (Döll et al., 2016)"

Will change to: "However, further advancements are needed to improve the integration of anthropogenic
impacts into hydrological models (Nazemi and Wheater, 2015a, b; Döll et al., 2016; Pokhrel et al.,
2016)"

*"The allocation of sectoral water demand to surface and ground-water systems as well as the sectoral*
*return flow into the surface water system seems to be equivalent to (Voisin et al., 2017), which should*
*then be cited."*

We have included the citation mentioned by the referee, as well as other studies (Hanasaki et al., 2018)
that used the same approach in allocation sectoral water demands to surface and groundwater systems.

Line 150-153: "The partitioning of water withdrawals between surface and ground water resources was based on the study of Döll et al. (2012), who estimated the groundwater withdrawal fraction for each sector in around 15.000 national and sub-national administrative units."

Will change to: "The partitioning of water withdrawals between surface and ground water resources is data driven, similar to other studies (e.g. Döll et al., 2012; Voisin et al., 2017; Hanasaki et al., 2018).

Groundwater withdrawal fraction were based on the study of Döll et al. (2012), who estimate fractions for each sector in around 15.000 national and sub-national administrative units."

*"The description of how the supply is allocated to the different sectoral water demands needs to be*

*specified in this manuscript. A missing description is how the priority is set between sectoral demands.*

*For example , are thermo-electric plants getting their demand met first before domestic or irrigation*

*demand?"*

The priority between sectoral water demands was described in section 2.2.1 (water withdrawal and consumption) on lines 162-163: "When water demands cannot be met, water withdrawals are allocated to the domestic, energy, manufacturing, livestock and irrigation sector in that order". However, we will make this more clear.

Lines 162-163: "When water demands cannot be met, water withdrawals are allocated to the domestic, energy, manufacturing, livestock and irrigation sector in that order"

Will change to: "In terms of water allocation, under conditions where water demands cannot be met, water withdrawals are allocated to the domestic, energy, manufacturing, livestock and irrigation sector in that order"

*"Is there any priority for supply allocation based on spatial location? Which grid cells can request*

*water from a mainstream if the main channel is not within this grid cell?"*

There is no priority for supply allocation based on location, inside or outside the delta. Water requests from the mainstream (if the main channel is not within the grid cell) are allocated based on demand.

This will be explicitly stated.

Line 159-160: "Therefore, streamflow at the river mouth is available for use in delta areas to simulate the actual water availability."

Will change to: "Therefore, streamflow at the river mouth is available for use in delta areas (partitioned based on demand) to simulate the actual water availability."

*"Was the Hanasaki et al. (2006) "dependence" database used?"*

The Hanasaki et al. (2006) dependence method is not used in this study, which will be explicitly stated.
Rather our study used the controlled discharge fraction as the fraction of downstream demands taken
into account. This is described in section 7.3 (appendix c: dam operation scheme) on lines 566-567:
"Water demands were based on the water demands of downstream cells. Only a fraction of water
demands were taken into account, based on the fraction of upstream area the dam controlled". However,
there was an error which causes confusion; "upstream area" should read "upstream discharge".

Line 566-567: "Only a fraction of water demands were taken into account, based on the fraction of
upstream area the dam controlled."

Will change to: "Only a fraction of water demands were taken into account, based on the fraction of
upstream discharge the dam controlled."

*"While authors indicate that it will be the subject of further research, what is the default implementation*
*that was used in the presented simulations?"*

We are not fully sure if we understand the referee correctly. However, assume the referee is referring to
which modules were used to generate the results in this study. We will explicitly add this information
to section 3.1 (setup).

Line 299: "(...) soil layers per grid cell. Soil and (natural) vegetation (...)"

Will change to: "(...) soil layers per grid cell. The routing, reservoir, irrigation and water-use modules
were all used in the simulations. The environmental flow requirements were only used where this is
specifically indicated. Soil and (natural) vegetation (...)"

*"the introduction is missing a range of large scale studies where such a large scale water management*
*model has been used with the VIC model, albeit not VIC5. While the proposed set up seems more*
*complete, it seems that the paper should still cite those studies as they represent to a certain extent an*
*earlier version of this integrated model (Voisin et al., 2017; Voisin et al., 2018; Zhou, Voisin, Leng,*
*Huang, & Kraucunas, 2018)"*

We have included almost all of the citations mentioned by the referee as they represent a wider range of
VIC model applications. Voisin et al. (2017) was excluded since this study seems to use the Community
Land Model (CLM) instead of the Variable Infiltration Capacity model (VIC).

Lines 80-84: "VIC has been used extensively in studies ranging from: coupled regional climate model
simulations (Zhu et al., 2009; Hamman et al., 2016), combined river discharge and water-temperature
simulations (van Vliet et al., 2016), hydrological sensitivity to climate change (Hamlet and Lettenmaier,

[revised manuscript text omitted]

---

## Author Comment (AC3) · 11 Mar 2020

Dear referee,

Thank you very much for reviewing our paper titled "Simulating human water impacts on global water resources using VIC-5" and for your valuable comments and suggestions. Below we address your comments (shown in italic), with our responses in blue.

**Model performance**

The referee suggests that we should further evaluate model performance *"compared to observed*

*sectoral and/or global water withrdrawals".* These suggestions were also addressed by the other reviewers.

We agree with these suggestions and we will include a rigorous evaluation of the hydrological model performance. We will compare model simulations with observations and/or reported data on discharge, total water storage, reservoir storage and sectoral water demands. As included in the response to the other reviewers, the following approaches are proposed:

1.  Simulated discharge will be compared with monthly timeseries and multi-year average discharge from the GRDC dataset, between 1980 and 2010. Stations are selected within the major river basins of the original VIC calibration paper of Nijssen et al. (2001). Naturalized discharge as well as human-modified discharge simulations will be compared in this manner.

2.  Simulated total water storage will be compared with monthly timeseries, multi-year-average total water storage and inter-annual water storage trends from the GRACE satellite dataset, for the period 2004-2016. To do so, a 300km gaussian filter will be applied to the simulated total water storage, as it is in the GRACE dataset. Total water storage will be compared for the same river basins as in the discharge comparison. Naturalized and human-modified total water storage simulations will be compared in this manner. These results will also include the  unmet water demands, subsequent non-renewable groundwater abstractions and long-term total water storage exploitation.

3.  Simulated sectoral water demand will be compared with monthly timeseries from the Huang et al. (2018) dataset. This is in addition to the comparison to the Shiklomanov (2000) dataset and

FAOSTAT (FAO, 2016), EUROSTAT (EC, 2019) and WWDR (Connor, 2015) datasets already used in the paper. Sectoral water demands will be compared for the world and for the 5 regions used in this paper (Africa, Americas, Asia, Europe and Oceania); and separately for each sector (irrigation, domestic, industrial and livestock) separately.

4.   Simulated reservoir inflow, storage and release will be compared with monthly timeseries from
 Yassin et al. (2019) (assuming this data is shared), Rougé et al. (2019) and Hanasaki et al. (2006)
 datasets. Dams are selected based on data availability and evaluation will focus on large dams.

**Novelty**

The referee comments that the *"methodology itself lacks in novel advancements"* and, in the specific
comments, that *"It should be more carefully noted throughout the text the novelty of what is being added*
*to the modeling community"*. Claims regarding its use in modelling the water-food-energy nexus *"may*
*be misleading"* and, in the specific comments, that such conclusions *"should be clarified"*. This was
also commented by another reviewer.

With regard to the notions of methodological novelty: we agree that the incorporated modules are based
on previous major works. However, the integration of these modules is a clear improvement compared
to previous VIC studies. Our model study includes the full range of water-use sectors (including
domestic, industrial, energy and livestock), which have been estimated independently. Also, the routing
module was fully integrated in VIC-5, which was not possible in previous VIC versions. This heavily
decreases computation times for human-impact studies and provides a much improved framework for
other future human-impact studies. Water-use sectors can also use groundwater as a resources, which
directly impacts baseflow and thus downstream (dry-season) water availability.

With regard to the notions of the water-food-energy nexus: we agree with the referee that notions
towards the modelling of the water-food-energy nexus may be misleading. We will therefore remove
these sentences from the manuscript, and rewrite part of the discussion.

For a full description of all proposed changes we refer to our responses to referee 1.

**Specific comments**

*"Line 328: the study is mentioned to use varying socioeconomic predictors. These could be better*
*explained in section 2.3.2 in order to specify where GDP and GVA are obtained."*

We will add an explanation to section 2.3.2, based on section 7.4.1.

Lines 243-244: "Domestic and industrial water withdrawals were estimated based on Gross Domestic
Product (GDP) per capita and Gross Value Added (GVA) by industries respectively."

Will change to: "Domestic and industrial water withdrawals were estimated based on Gross Domestic
Product (GDP) per capita and Gross Value Added (GVA) by industries respectively (from Bolt et al.
(2018), Feenstra et al. (2015) and World bank (2010); see section 7.4.1 for more details)."

*"Lines 406-408: "To our knowledge no previous study has estimated the amount of*

*global non-renewable groundwater withdrawals without using on the the models mentioned*

*above" - see Turner et al. (2019) or Kim et al. (2016) for additional groundwater*

*withdrawal modeling capabilities."*

We thank the referee for these useful citations, which we will incorporate into the text.

*"Line 426: "Note that VIC-WUR does not include non-renewable groundwater withdrawals,*

*while these withdrawals would affect baseflow to a lesser degree" - I am confused,*

*then why was there a discussion on about this in paragraph starting at line 400?*

*Maybe consider reorganizing these thoughts.."*

The discussion in the paragraph starting at line 400 assumes that all unmet water withdrawals originate from non-renewable sources. However, this does not mean that models actually include simulations of non-renewable groundwater withdrawals. To make this distinction clearer we will include more detail about the model setup used in the results, and we will reorganize the discussion.

We hope the referee agrees with the changes made, and are open to any further suggestions or comments.

Sincerely,

Bram Droppers on behalf of all co-authors

**References**

Bolt, J., Inklaar, R., de Jong, H., and van Zanden, J. L.: Rebasing 'Maddison': New income comparisons
and the shape of long-run economic developments, University of Groningen, Groningen, the
Netherlands, 69, 2018.

Connor, R.: Water for a sustainable world, United Nations Educational, Scientific and Cultural
Organisation, Paris, France, 139, 2015.

Feenstra, R. C., Inklaar, R., and Timmer, M. P.: The Next Generation of the Penn World Table, Am
Econ Rev, 105, 3150-3182, 10.1257/aer.20130954, 2015.

Huang, Z., Hejazi, M., Li, X., Tang, Q., Vernon, C., Leng, G., Liu, Y., Döll, P., Eisner, S., Gerten, D.,
Hanasaki, N., and Wada, Y.: Reconstruction of global gridded monthly sectoral water withdrawals for
1971–2010 and analysis of their spatiotemporal patterns, Hydrol. Earth Syst. Sci., 22, 2117-2133,
10.5194/hess-22-2117-2018, 2018.

Shiklomanov, I. A.: Appraisal and assessment of world water resources, Water Int, 25, 11-32, Doi
10.1080/02508060008686794, 2000.

---

## Author Response (AR2)

Dear editor,

Thank you very much for handling our manuscript again. Below we provided a list of all the relevant changes made in the manuscript. The referee responses and a mark-up of the new manuscript version are also included.

**Referee response**

Several textual changes have been made in response to referee comments and suggestions. This includes minor changes to the abstract and conclusions and changes to the results section.

**Other changes**

1. Figure 3 caption was wrong and has been adjusted
2. Figure 9 outlining was improved
3. Code and documentation availability section was updated with new DOI's
4. Removed some double spaces

We hope this list (and attached referee responses and manuscript mark-up) sufficiently describes the manuscript changes made.

Sincerely,

Bram Droppers on behalf of all co-authors

**Referee 1 - Author response**

Dear referee,

Thank you very much for reviewing our paper again. We are pleased that our adjustments the manuscript have been received favourably. Below we address your comments and suggestions (shown in italic), with our responses in blue.

**Motivations for irrigation efficiency**

"*I have just noticed that the authors may have misunderstood one of my comments (...) I wanted to point out that e* [irrigation efficiency] *would work only if the authors' estimations of C* [consumptive water ues] *were by chance quite similar to that of Frenken and Gillet (2012). I know this is a question without answer, but at least, if the authors have any concrete idea why they chose Frenken and Gillet (2012) it should be worth mentioned here*."

We agree with the referee that the irrigation efficiency is heavily dependent on the simulated irrigation water consumption (crop evapotranspiration). There have been various other assessments of irrigation efficiency, which all used different methods and models (e.g. Döll and Siebert, 2002; Rohwer et al., 2007; Jägermeyr et al., 2015)

We decided to select Frenken and Gillet (2012) since it is a relatively recent and comprehensive study with a high resolution crop modelling basis (5 by 5 arc-minute spatial resolution). Information regarding crop growing areas and seasons were gathered from the same database as the reported crop irrigation water withdrawal, making these values comparable and consistent (to an extent). Therefore, we found this study to be suitable in estimating the actual irrigation efficiency on a aggregated scale.

Lines 255-260: "The water loss fraction was based on Frenken and Gillet (2012), who estimated the irrigation efficiency for 22 United Nations sub-regions. Irrigation efficiencies were estimated based on the differences between the calculated crop water requirements (crop evapotranspiration; consumptive water use) and the reported irrigation water withdrawals (including transportation and application losses). Crop water requirements are estimated based on the FAO Irrigation and Drainage paper (Allen et al., 1998). Low irrigation efficiencies can result in irrigation water withdrawals up to four times higher than the crop water requirements in regions such as east- and west Africa."

Will change to: "The water loss fraction was based on Frenken and Gillet (2012), who estimated the aggregated irrigation efficiency for 22 United Nations sub-regions. Irrigation efficiencies were estimated based on the difference between AQUASTAT reported irrigation water withdrawals and calculated irrigation water requirements (Allen et al., 1998), using data on crop information (e.g.

growing season, harvest area) from AQUASTAT."

We hope the referee agrees with our response, and are open to any further suggestions or comments.

Sincerely,

Bram Droppers on behalf of all co-authors

While we will include further recommendations for upcoming for the sectoral water withdrawal setup,
the manuscript will mostly refrain from any commenting on upcoming energy-water-land analytics. We
think that the model setup as presented currently requires further development for integrated energy-
water-land analytics. For example, while the current model setup is able to estimate the effects of
anthropogenic water use on water availability (and water stress), this is not translated to actual impacts.
This is currently our primary focus.

Further efforts are also needed to better represent the energy sector, especially for energy-water-land
analysis. Currently only a small part of the energy sector is included, while this is an important sector
in the energy-water-land nexus. Lastly, as the irrigation sector is integrated into the hydrological model,
which will be part continuous model development.

Lines 510-511: "However, note that the model setup of VIC-WUR allows for the evaluation of other
sectoral water demand inputs, on various temporal aggregations."

Will change to: "While the current setup to estimate sectoral water demands is well suited for future
water withdrawal estimations, there are various other approaches (e.g. Alcamo et al., 2003; Vassolo and
Döll, 2005; Shen et al., 2008; Hanasaki et al., 2013; Wada and Bierkens, 2014). As the model setup of

VIC-WUR allows for the evaluation of other sectoral water demand inputs (on various temporal
aggregations), several different approaches can be used depending on the focus region and data-
availability for calibration."

Line 378: "(...) was limited before 1990."

Will change to: "(...) was limited before 1990. Also, data on the disaggregation of industrial sectors (e.g.
energy and mining) was limited, which can be important sectors in the water-food-energy nexus."

We hope the referee agrees with the changes made, and are open to any further suggestions or comments.

Sincerely,

Bram Droppers on behalf of all co-authors

| LPJmL | 2555 | 1971 - 2000 | Rost et al. (2008) |
| PCR-GLOB | (a) 2644
(b) 2309 \*\* | (a) 2010
(b) 2000 - 2015 | (a) Wada and Bierkens (2014)
(b) Sutanudjaja et al. (2018) |
| WaterGAP | (a) 3185
(b) 2400 | (a) 1998-2002
(b) 2003 - 2009 | (a) Döll et al. (2012)
(b) Döll et al. (2014) |
| WBM | 2997 | 2002 | Wisser et al. (2010b) |

**Commented [DB10]:** Updated references

[revised manuscript text omitted]

| | | | | |
|---------|------------|-----------|---------------|---------------------------|
| VIC-WUR | 992 (± 51) | 316 (± 63) | 1980 - 2016 | Our study |
| H08 | 789 (± 30) | 182 (± 26) | 1984 - 2013 | Hanasaki et al. (2018) |
| MATSIRO | 570 (± 61) | 330 | 1998 - 2002 | Pokhrel et al. (2015) |
| GCAM | | (a) 600
(b) 550 | (a) 2005
(b) 2000 | (a) Kim et al. (2016)
(b) Turner et al. (2019) |
| PCR-GLOB | (a) 952
(b) 632 | (a) 304
(b) 171 | (a) 2010
(b) 2000 - 2015 | (a) Wada and Bierkens (2014)
(b) Sutanudjaja et al. (2018) |
| WaterGAP | (a) 1519
(b) 888 | (a) 250
(b) 113 | (a) 1998-2002
(b) 2000 - 2009 | (a) Döll et al. (2012)

[revised manuscript text omitted]

---

## Author Response (AR3)

Dear editor,

Thank you very much for your efforts in the review process. We are happy to receive such good news!

We have made some minor changes according to the manuscript preparation requests of GMD, such as: Changing "Figure" to "Fig.", changing "Section" to "Sect.", embedding fonts into the figures, changed the table design to only have horizontal lines, double-checked the references and other minor changes. Lastly we added to units to the RMSE in the abstract as requested.

We hope these details (and attached manuscript mark-up) sufficiently describes the manuscript changes made. We are looking forward to the proof version of our manuscript.

Sincerely,

Bram Droppers on behalf of all co-authors

[revised manuscript text omitted]